# Microbial community composition across a coastal hydrological system affected by submarine groundwater discharge (SGD)

**Dini Adyasari**[1]*, **Christiane Hassenrück**[1], **Daniel Montiel**[2,3], **Natasha Dimova**[2]

**1** Department of Biogeochemistry and Geology, Leibniz Centre for Tropical Marine Research, Bremen, Germany, **2** Department of Geological Sciences, Coastal Hydrogeology Laboratory, University of Alabama, Alabama, AL, United States of America, **3** Geosyntec Consultants, Clearwater, FL, United States of America

* dini.adyasari@leibniz-zmt.de

## Abstract

Mobile Bay, the fourth largest estuary in the USA located in the northern Gulf of Mexico, is known for extreme hypoxia in the water column during dry season caused by $NH_4^+$-rich and anoxic submarine groundwater discharge (SGD). Nutrient dynamics in the coastal ecosystem point to potentially elevated microbial activities; however, little is known about microbial community composition and their functional roles in this area. In this study, we investigated microbial community composition, distribution, and metabolic prediction along the coastal hydrological compartment of Mobile Bay using 16S rRNA gene sequencing. We collected microbial samples from surface (river and bay water) and subsurface water (groundwater and coastal pore water from two SGD sites with peat and sandy lithology, respectively). Salinity was identified as the primary factor affecting the distribution of microbial communities across surface water samples, while DON and $PO_4^{3-}$ were the major predictor of community shift within subsurface water samples. Higher microbial diversity was found in coastal pore water in comparison to surface water samples. *Gammaproteobacteria*, *Bacteroidia*, and *Oxyphotobacteria* dominated the bacterial community. Among the archaea, methanogens were prevalent in the peat-dominated SGD site, while the sandy SGD site was characterized by a higher proportion of ammonia-oxidizing archaea. *Cyanobium PCC-6307* and unclassified *Thermodesulfovibrionia* were identified as dominant taxa strongly associated with trends in environmental parameters in surface and subsurface samples, respectively. Microbial communities found in the groundwater and peat layer consisted of taxa known for denitrification and dissimilatory nitrate reduction to ammonium (DNRA). This finding suggested that microbial communities might also play a significant role in mediating nitrogen transformation in the SGD flow path and in affecting the chemical composition of SGD discharging to the water column. Given the ecological importance of microorganisms, further studies at higher taxonomic and functional resolution are needed to accurately predict chemical biotransformation processes along the coastal hydrological continuum, which influence water quality and environmental condition in Mobile Bay.

**Data Availability Statement:** Primer-clipped sequences from this study have been submitted to the European Nucleotide Archive (ENA) with the project accession number PRJEB33004.

Environmental data, OTU table, and its taxonomic affiliation are available in https://doi.org/10.1594/PANGAEA.912763.

**Funding:** DA is funded by DAAD Sustainable Water Management Grant No 57156376 and DFG Bernd Rendel Prize 2019. DM and ND are funded by National Science Foundation (NSF OIA-1632825), the 2016 ExxonMobil Summer Fund, the 2015 Gulf Coast Association of Geological Societies Student Research Grant, the University of Alabama Graduate School Research and Travel Support Fund, the UA Department of Geological Sciences W. Gary Hooks, and the A. S. Johnson Travel Fund. Geosyntec Consultant provided support in the form of salaries for DM, but did not have any additional role in the study design, data collection and analysis, decision to publish, or preparation of the manuscript. The other funders had no role in study design, data collection and analysis, decision to publish, or preparation of the manuscript. The specific roles of these authors are articulated in the 'author contributions' section.

**Competing interests:** DM and Geosyntec Consultants declare no competing interest regarding the publication of all information and data included in this manuscript. DM and Geosyntec Consultants declare that no employment, consultancy, patent, product development, or marketed products will be implemented in relation with this publication. DM's affiliation with Geosyntec Consultants does not alter our adherence to PLOS ONE policies on sharing data and materials.

## Introduction

Submarine groundwater discharge (SGD) is defined as groundwater flow across the land-ocean interface to the coastal water. SGD is known to transport chemical and biological constituents, such as nutrients [1–3], trace metals [4, 5], and bacteria [6, 7]. In an environment with high primary productivity, SGD is reported to cause eutrophication and harmful algal blooms (HAB) [8–10]. In some cases, SGD influences the composition and abundance of marine biota in the receiving coastal water, e.g. fish [11, 12], bacteria [13, 14], macrophytes [15, 16], or phytoplankton [17, 18].

Subterranean estuaries (STEs), where SGD flows through before discharging to the coastal water, are active mixing zones and biogeochemical cycling hot spots [19]. They are subjected to both seasonal water table fluctuation and increased nutrient input from the groundwater coming from the land side, as well as changes in oxygen saturation, quick redox switches and organic matter inputs from tidal fluctuation and sea level fluctuation from the marine side [20–22]. It has been found that STE could play a role as either a source [23] or sink of nutrient species [24, 25]. The attenuation or changes of speciation of nutrients is attributed to chemically and biologically mediated reactions. For examples, microbial communities are found to play a significant role in mediating nitrogen cycling in the STEs, such as nitrification and denitrification, and subsequently alter the chemical composition of SGD discharging to the overlying water [26–28]. Alternative microbially mediated pathways for nitrate reduction, such as dissimilatory nitrate reduction to ammonium (DNRA) and anaerobic ammonium oxidation (Annamox), were also observed in STEs [29, 30]. Archaea, while usually comprising a minor fraction of microbial communities in aquatic environments, have been shown to play an essential role in the carbon and nitrogen cycle, particularly related to nitrification as well as methane production and oxidation [31, 32]. The composition of microbial communities inhabiting the STE is governed by physicochemical factors, such as, but not limited to salinity [33], redox condition [34], temperature [35], or tidal fluctuation [36]. However, in comparison with surface estuaries, the understanding of the subsurface microbial community and their response to biotic and abiotic reactions is still limited.

Mobile Bay, a typical estuary in the northern Gulf of Mexico, is also the largest estuary east of the Mississippi Delta. Due to extreme hypoxia in the water column during dry seasons, large-scale fish and crustacean kills, locally known as *Jubilees* often occur in this part of northeast of the Gulf of Mexico [37–39]. More recent studies indicate that both the *Jubilees* and HABs occur at specific locations of Mobile Bay and are associated with areas without direct surface water inputs [40, 41]. Ultimately, SGD was investigated as one of the main contributors to the hypoxia occurring in the bay. Indeed, Montiel, Lamore [24] indicated that anoxic SGD delivered nearly a quarter of the total nutrient inputs to Mobile Bay during the dry season in the form of $NH_4^+$ and DON. More importantly, the significant SGD-derived N fluxes that occur exclusively to the east shore of Mobile Bay, whose coastal lithology is dominated by an underlying peat layer. Montiel, Lamore [24] suggested that the identified organic-rich sedimentary layer alters the composition of $NO_3^-$ dominated groundwater observed further inland into $NH_4^+$ and DON-prevalent SGD discharging to Mobile Bay.

Microbial investigations have been conducted in Mobile Bay before; however these studies were mostly related to surface water characterization [42–44], while studies on groundwater are still limited [30]. Given that Mobile Bay is one of the most developed estuaries in the northern Gulf of Mexico, and of the ecological importance of microbial communities for mediating coastal biogeochemical processes in areas with nutrient-rich groundwater, it is fundamental to better understand their distribution, diversity, and function. In this paper, we examined microbial community composition and distribution along the coastal hydrological continuum

of Mobile Bay affected by SGD using 16S rRNA gene sequencing. Microbial samples were collected (a) horizontally along the coastal hydrological continuum, from groundwater to the STE, river, and water column of the bay, and (b) along with vertical profiles at two SGD sites with contrasting hydrogeological properties and SGD regimes. In addition, functional profiles related to nitrogen cycling were predicted using Tax4Fun2 [45].

## Materials and methods

### Site description and sample collection

Mobile Bay has an area of 1.3 x 109 m$^2$, an average depth of 3.5 m, and a total volume of 4.6 x 10$^9$ m$^3$. During the dry season, when we conducted our sampling expedition, the average mean temperature and precipitation rate were 27˚C and 1670 mm, respectively. Although there was no reported hypoxia during this specific sampling campaign, sample collection was conducted during a period when anoxic events occur more frequently. Recent hydrogeological studies in Mobile Bay have revealed two major points of groundwater discharge at the southeast shore (which will be further abbreviated as TS-SE) and the northeast shore (which will be further abbreviated as TS-NE) of Mobile Bay [24, 41] (Fig 1). Montiel, Lamore [24] reported that at the TS-SE site, SGD is on average 0.15–0.25 m d$^{-1}$ annually, whereas the SGD-delivered $NH_4^+$, $NO_3^-$, DON, and $PO_4^{3-}$ for the four-year duration of their study were 11–35 mmol m$^{-2}$ d$^{-1}$, 1.2–5 mmol m$^{-2}$ d$^{-1}$, 9–32.5 mmol m$^{-2}$ d$^{-1}$, and 0–0.2 mmol m$^{-2}$ d$^{-1}$, respectively. Sediment cores recovered from the SGD hot spot consisted of a coarse beach sand layer of 0.5 m thickness and 3% organic matter content, underlain by a 1.5-m organic-rich black fine sand with an organic matter content of up to 36% (peat layer), which was in contact with the Miocene–Pliocene Aquifer with low organic matter content (Fig 1). For comparison, at TS-NE, SGD fluxes were 0.17–0.25 m d$^{-1}$, and $NH_4^+$, $NO_3^-$, DON, and $PO_4^{3-}$ fluxes were 1–2 mmol m$^{-2}$ d$^{-1}$, 2.2–7.5 mmol m$^{-2}$ d$^{-1}$, 2.7–8 mmol m$^{-2}$ d$^{-1}$, and 0–0.1 mmol m$^{-2}$ d$^{-1}$, respectively. At this second site, the piezometer was placed through the local STE where shallow groundwater percolates through layers consisting exclusively of coarse sand without an indication of vertical structuring.

We declare that the research team who carried the field work in Mobile Bay, Alabama, USA did not need permission for accessing the field site as described. The groundwater wells sampled for this study were private properties and at the time of sampling the owners were present and gave personally access for groundwater collection. In total, we collected 20 samples across the coastal hydrological land-aquifer-ocean continuum of Mobile Bay (Fig 1): one sample of inland groundwater (GW-1); one sample (TS-SE-A) from a 2 m-depth piezometer at the TS-SE site; five pore water samples from a vertical profile collected in 0.5 m intervals from a 2.5 m-deep piezometer that is located in 3 m distance from TS-SE-A (TS-SE-B1, TS-SE-B2, TS-SE-B3, TS-SE-B4, TS-SE-B5); two pore water samples in a vertical profile taken in 1.5 m intervals from a 3 m-depth piezometer at the TS-NE site (TS-NE-A1, TS-NE-A2); one sample (TS-NE-B) from a 2 m-deep piezometer located 15 m from TS-NE-A; three river samples from the three main tributaries entering the bay (Mobile, Apalachee, and Blakeley rivers, abbreviated as MR, AR, BR, respectively), and Mobile Bay water column samples (MB1-7). Pore water from SGD sites was collected from piezometers located in the intertidal areas, and all samples were taken during the low tide when the SGD signal is the most pronounced. Samples from the Mobile Bay water column were collected from different parts of the bay to cover a gradient of salinity and river influence: samples MB1-4 were collected in the southern part of Mobile Bay where the river influence is almost negligible, while MB6-7 were taken close to estuaries in the northern part of the bay. The water sample for GW-1 was collected after pumping the well with a submersible pump at a constant rate until conductivity, temperature, and DO values were stable. Water samples from Mobile Bay and its tributaries were collected during boat

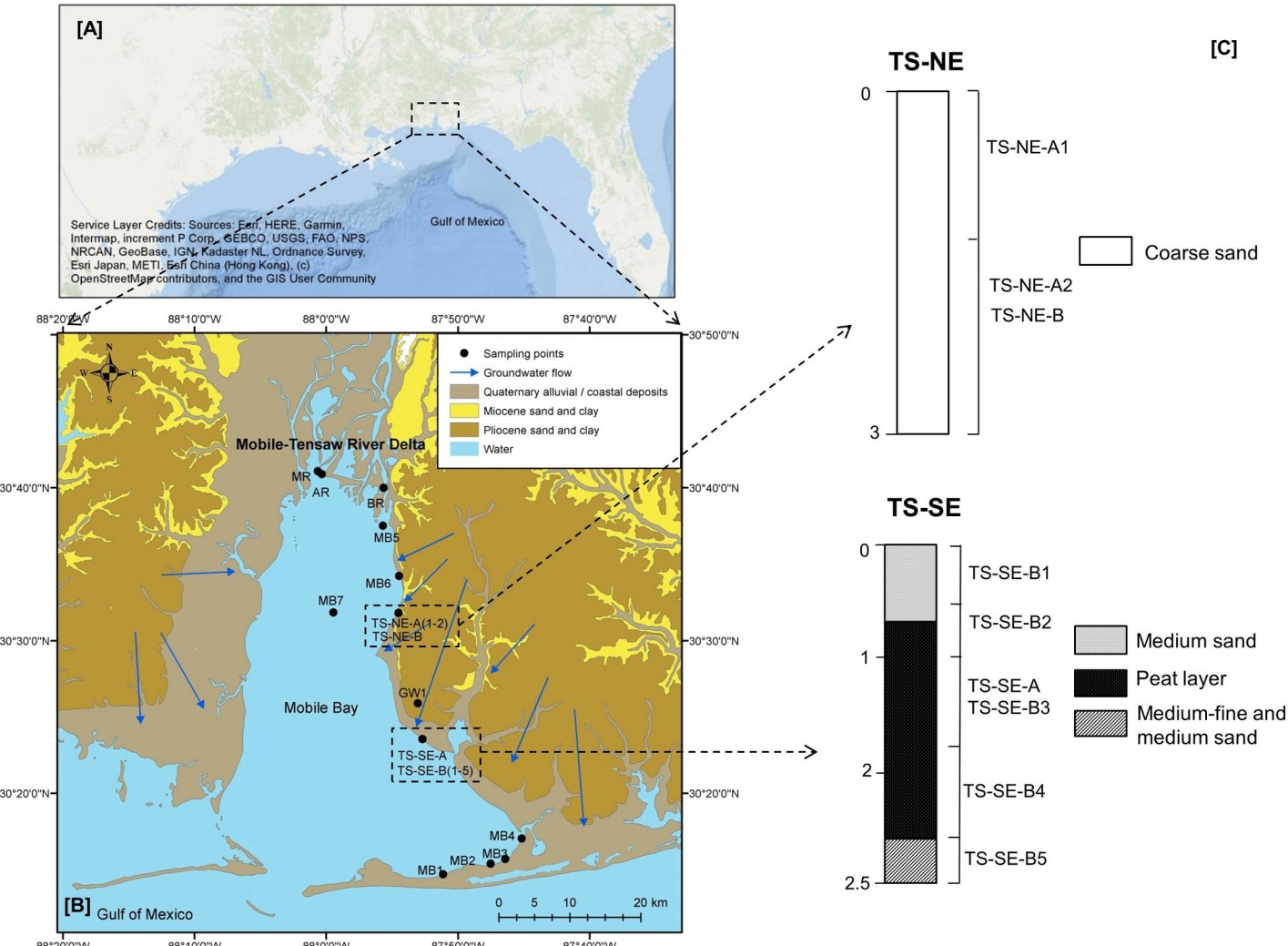

**Fig 1.** Study site (A) and sampling points (B) in Mobile Bay, USA, and vertical lithological section of sediment cores from TS-NE and TS-SE (C). The geological map is modified from a previous version prepared at the Geological Survey of Alabama [46], while groundwater flow and lithology profile are adapted from Montiel, Lamore (24). TS-NE-B and TS-SE-A were sampled from different piezometer but similar depth with TS-NE-A2 and TS-SE-B3, respectively.

surveys with a submersible pump from a depth of 0.3 m. Inland groundwater (GW-1) and pore water from TS-SE and TS-NE are categorized as subsurface samples, while river and Mobile Bay water column samples are grouped as surface water samples.

For microbial analysis, 250 ml of water samples were collected, filtered through a 0.2 μm isopore membrane filter, and stored frozen in a sterile microcentrifuge tube until analysis. Physical water parameters (salinity, temperature, and dissolved oxygen (DO)) and samples for nutrient analysis ($NO_3^-$, $NH_4^+$, DON and $PO_4^{3-}$) were also collected and measured alongside the microbial samples. The physical parameters were recorded with a Pro2030 (YSI Inc.) handheld instrument with a Galvanic sensor. Nutrient samples were filtered in the field through sterile 0.45 μm cellulose acetate filters and stored in acid-cleaned 50 mL polypropylene vials. Samples were kept on ice for a maximum of 6 hours until they were frozen pending analysis. The analyses were performed at the Dauphin Island Sea Lab (DISL) using a Skalar San$^{++}$ segmented flow autoanalyzer with automatic in-line sample digestion (Skalar Analytical B.V).

## Microbial community analysis

The DNA extraction of the microbial samples was conducted at the Leibniz Centre for Tropical Marine Research (ZMT), Germany, based on procedures described in Nercessian, Noyes [47]. The Primer set 515YF (5′-GTGYCAGCMGCCGCGGTAA-3′)/926R(5′-CCGYCAATT YMTTTRAGTTT-3′) [48] was used to obtain DNA sequences from the V4-V5 hypervariable region of the 16S rRNA gene. Amplicon sequencing was implemented on the Illumina Miseq platform at LGC Genomics (Berlin). Bioinformatic sequence processing was conducted using the DADA2 package [49]. Forward and reverse reads were quality trimmed to 215 bp at a maximum expected error rate of 4. Error learning and denoising were conducted using all sequences of the data set and pooling sequences across all samples. Forward and reverse reads were merged using default parameters and chimeras were removed with the method 'consensus'. Only amplicon sequence variants between 361 and 399 bp occurring at least twice in the data set were retained and will further be referred to as operational taxonomic units (OTUs). OTUs were taxonomically classified using the silvangs web service (https://www.arb-silva.de/ngs/, date accessed 23.08.2018) with version 132 of the SILVA reference database. Unwanted lineages (chloroplasts, mitochondria) and OTUs unclassified on the phylum level were removed from the dataset.

The prediction of functional genes based on 16S rRNA gene sequences was conducted using Tax4Fun2 [45], where OTUs were associated with specific KEGG orthologue functional genes (KO) and their subsequent empirical metabolic pathways. As SGD studies suggested $NH_4^+$-rich groundwater discharging from the eastern shore of Mobile Bay, KOs related to nitrogen metabolism (i.e. nitrification, denitrification, DNRA, assimilatory nitrate reduction, and nitrogen fixation) were chosen for further analysis and discussion. As methanotrophic and methanogenic communities were found in the study site, KOs related to the methane cycle were likewise investigated.

Primer-clipped sequences from this study have been submitted to the European Nucleotide Archive (ENA) with the project accession number PRJEB33004, using the data brokerage service of German Federation for Biological Data [50]. The OTU table and taxonomic classifications are available online (https://doi.pangaea.de/10.1594/PANGAEA.912763).

## Statistical analysis

Statistical analyses were implemented in R version 3.5.2 [51] using the vegan package version 2.5–6 [52]. Number of OTUs, Shannon, and inverse Simpson indices were calculated to estimate the alpha diversity of samples with more than 10000 sequences, randomly subsampling the data set to this sequencing depth 100 times. Beta diversity was assessed based on Bray-Curtis (BC) dissimilarities of relative sequence proportions and visualized using complete linkage hierarchical clustering for all microbial samples.

Prior to any statistical analyses, samples were divided into surface and subsurface samples because the relationships between environmental parameter in surface and subsurface samples were fundamentally different. Principal component analysis (PCA) was used to cluster sampling sites based on observed environmental parameters in surface and subsurface water samples. Missing data among the subsurface samples for the parameters salinity, temperature, and DO were estimated based on observations from a previous sampling expedition at the same sampling location. Non-metric multidimensional scaling (NMDS) was employed as ordination method for surface and subsurface microbial community composition. Environmental parameters were mapped to the NMDS plot using *envfit*, and the 3% most frequent and the 90% best fitted OTUs to environmental parameter were displayed in the NMDS ordination using the function *ordiselect* of the goeveg package version 0.4.2 [53]. Permutational multivariate

analysis of variance (PERMANOVA) using the function *adonis2* was applied to relative OTU proportions of surface and subsurface microbial samples to determine the contribution of environmental parameters in explaining the variation in microbial community composition. The best model was determined by forward model selection based on a significant ($p < 0.1$) increase in explained variation ($R^2$). Due to the unavailability of salinity, DO, and temperature data for some of TS-SE and TS-NE samples, these parameters were excluded from the selection of environmental predictors for the subsurface communities. Due to a high collinearity, $PO_4^{3-}$, DON, and $NO_3^-$ were used to approximate the effects of salinity, DO, and temperature, respectively ($PO_4^{3-}$ and salinity: Pearson r = 0.98, DON and DO: Pearson r = -0.74, $NO_3^-$ and temperature: Pearson r = -0.63).

## Results

### Environmental characteristics

All measurements of environmental parameters of the collected water samples are listed in Table 1. In the vertical profile of TS-SE, salinity increased with depth in the STE, while the salinity profile in TS-NE was uniform. Salinity also varied laterally within the bay. The southern part of the bay that was the furthest from the river entries had higher salinity (3.9–5.5 at MB1-4), whereas the northern part at the mouth of the rivers had lower salinity in the same range as the river (0.1–0.4 at MB5-7). During this sampling campaign, all surface water samples were well-oxygenated (DO > 2 mg L$^{-1}$), while groundwater and pore water samples were hypoxic (DO < 1 mg L$^{-1}$). $NO_3^-$ concentrations varied across all samples, ranging from 0–13.6 µM in surface water samples, and 0–155.9 µM in groundwater and pore water samples.

**Table 1. Physicochemical parameters at the study sites.**

| Sample | Longitude | Latitude | Salinity | DO (mg L$^{-1}$) | Temperature (°C) | $NO_3^-$ (µM) | $NH_4^+$ (µM) | $PO_4^{3-}$ (µM) | DON (µM) |
|--------|-----------|----------|----------|------------------|------------------|---------------|---------------|------------------|----------|
| GW-1 | -87.88392 | 30.4314 | 0.0 | 0.3 | 21.1 | 155.9 | 0.8 | 0.1 | 83.0 |
| TS-SE-A | -87.87815 | 30.39243 | 0.0 | 0.1 | 29.3 | 0.7 | 125.2 | 0.1 | 98.0 |
| TS-SE-B1 | -87.87825 | 30.39229 | 0.0 | 0.6 | 29.9 | 81.9 | 54.5 | 0.6 | 33.0 |
| TS-SE-B2 | -87.87825 | 30.39229 | 1.8 | 0.6* | 29.4 | 63.9 | 4.5 | 2.7 | 32.0 |
| TS-SE-B3 | -87.87825 | 30.39229 | 1.8 | 0.0* | 29.4* | 3.3 | 119.3 | 2.3 | 57.0 |
| TS-SE-B4 | -87.87825 | 30.39229 | 1.8* | 0.0* | 29.4* | 2.8 | 92.0 | 1.8 | 60.0 |
| TS-SE-B5 | -87.87825 | 30.39229 | 1.8* | 0.0* | 29.4* | 2.3 | 51.8 | 0.6 | 92.0 |
| TS-NE-A1 | -87.90843 | 30.53017 | 0.1 | 1.3 | 25.5 | 78.0 | 0.5 | 0.2 | 10.0 |
| TS-NE-A2 | -87.90843 | 30.53017 | 0.1* | 1.4* | 25.5 | 145.0 | 3.9 | 0.5 | 4.0 |
| TS-NE-B | -87.90863 | 30.53016 | 0.0 | 0.4 | 26.4 | 3.1 | 16.9 | 0.1 | 13.0 |
| MB1 | -87.85198 | 30.24455 | 3.9 | 3.6 | 27.9 | 0.9 | 0.4 | 0.2 | 42.1 |
| MB2 | -87.79181 | 30.25622 | 4.4 | 3.8 | 29 | 0.7 | 0.0 | 0.1 | 33.3 |
| MB3 | -87.75262 | 30.28377 | 5.5 | 4.6 | 30.1 | 0.7 | 0.2 | 0.1 | 32.5 |
| MB4 | -87.77329 | 30.26144 | 4.9 | 3.2 | 29.1 | 0.7 | 0.3 | 0.2 | 27.7 |
| MB5 | -87.92827 | 30.62521 | 0.1 | 3.8 | 24.8 | 13.6 | 1.3 | 0.6 | 44.0 |
| MB6 | -87.90781 | 30.57037 | 0.4 | 2.1 | 25.1 | 4.8 | 0.3 | 0.6 | 36.9 |
| MB7 | -87.99113 | 30.53048 | 0.1 | 3.4 | 29.5 | 1.4 | 0.3 | 0.9 | 35.7 |
| MR | -88.01071 | 30.68467 | 0.1 | 2.6 | 29.4 | 9.3 | 1.2 | 0.5 | 33.5 |
| BR | -87.92734 | 30.66650 | 0.1 | 3.5 | 29.4 | 7.6 | 0.8 | 0.4 | 30.8 |
| AR | -88.00559 | 30.68133 | 0.0 | 3.5 | 27.5 | 11.5 | 1.5 | 0.6 | 32.7 |

Asterisks (*) indicate recalculated values obtained from Montiel, Lamore (24) based on patterns from a previous sampling campaign at the same sampling location as no observations were available during our expedition.

The highest $NO_3^-$ concentration (155 µM) was found in the inland well, which was located 2.5 km from the coastline. Low $NH_4^+$ concentration (0–1.4 µM) in surface water contrasted with high $NH_4^+$ concentration in pore water, whereas the highest $NH_4^+$ concentrations and the highest range of $NH_4^+$ concentration was found in the vertical profile of TS-SE-B (4–125 µM). $PO_4^{3-}$ concentrations ranged between of 0 and 2 µM across all samples. DON concentration displayed a higher range in the subsurface (4–98 µM) compared to surface water samples (27–45 µM). Between two SGD sites, TS-SE had a considerably elevated concentration of DON (32–98 µM) in comparison to TS-NE (4–13 µM).

The PCA for environmental parameters in surface water showed that PC1, which accounted for 52.2% of the variation in the data, separated MB1-4 from the rest of the surface water samples and was mainly associated with DO, $PO_4^{3-}$, DON, and $NH_4^+$ concentrations, while PC2 was mainly determined by temperature, salinity, and $NO_3^-$ (Fig 2A). Different patterns of association between environmental variables were found in subsurface samples, where 55.6% of the variability of subsurface environmental parameters was explained by PC1, which separated TS-NE and the groundwater sample from the samples from TS-SE (Fig 2B). Among subsurface samples, DO and $NO_3^-$ concentration were negatively associated with salinity, temperature, $PO_4^{3-}$, DON, and $NH_4^+$.

### Microbial diversity and community ordination

In total, 565,793 sequences represented in 7827 OTUs were obtained for further analysis with an average of 28,289 sequences per sample. We identified 7687 bacterial OTUs contained in

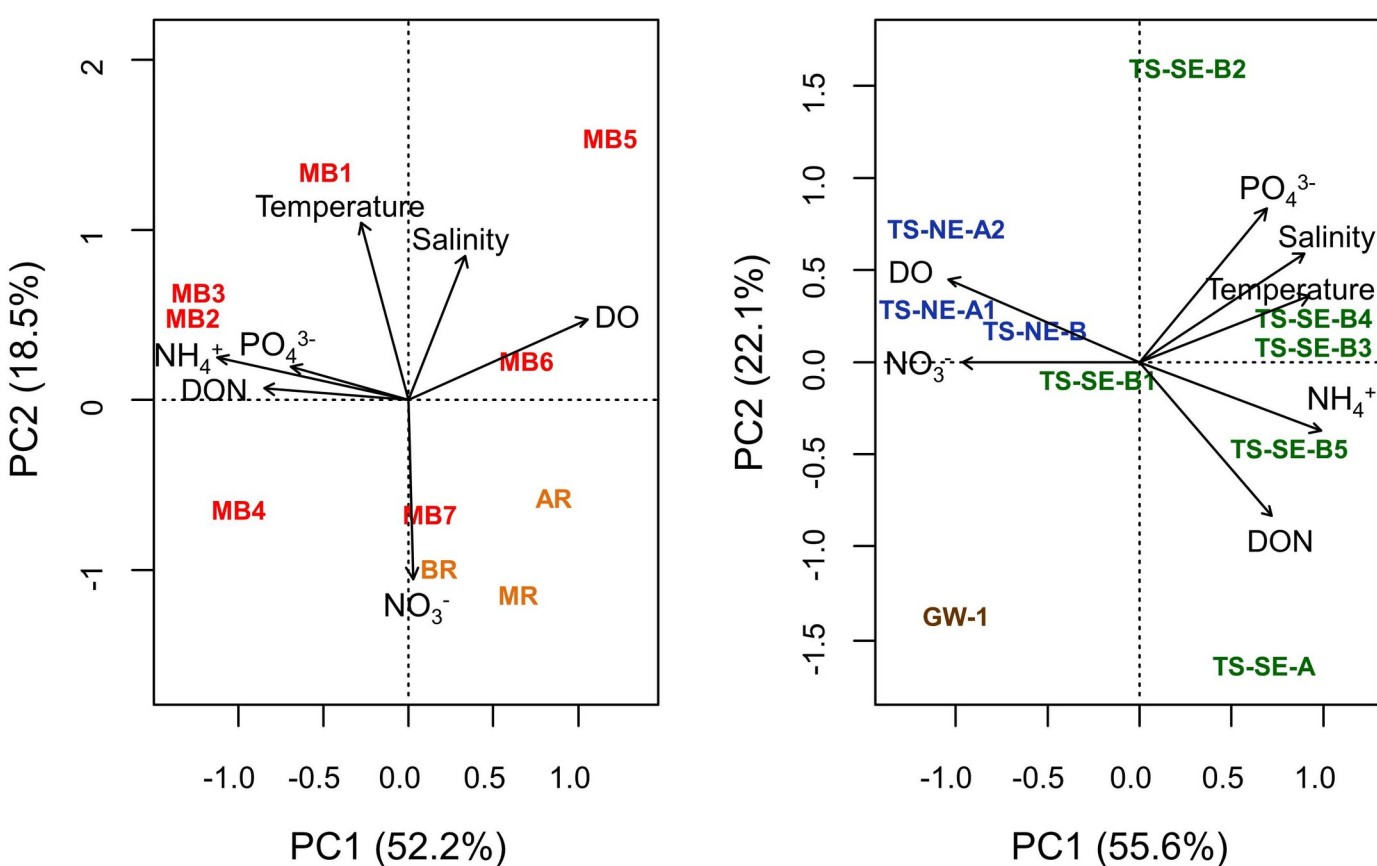

**Fig 2.** Principal Component Analysis (PCA) for based on the environmental parameters for surface water (A) and subsurface water (B). Nitrate: $NO_3^-$; Dissolved organic nitrogen: DON; Ammonium: $NH_4^+$; Phosphate: $PO_4^{3-}$; Dissolved oxygen: DO.

558,435 sequences and 140 archaeal OTUs represented in 7358 sequences. There were three samples with less than 10,000 sequences (i.e. TS-SE-B4, TS-NE-A1, and MB6), which we excluded from the alpha diversity analysis due to insufficient sequencing depth. We observed trends among the three investigated alpha diversity indices (Fig 3), where TS-SE-B2 and

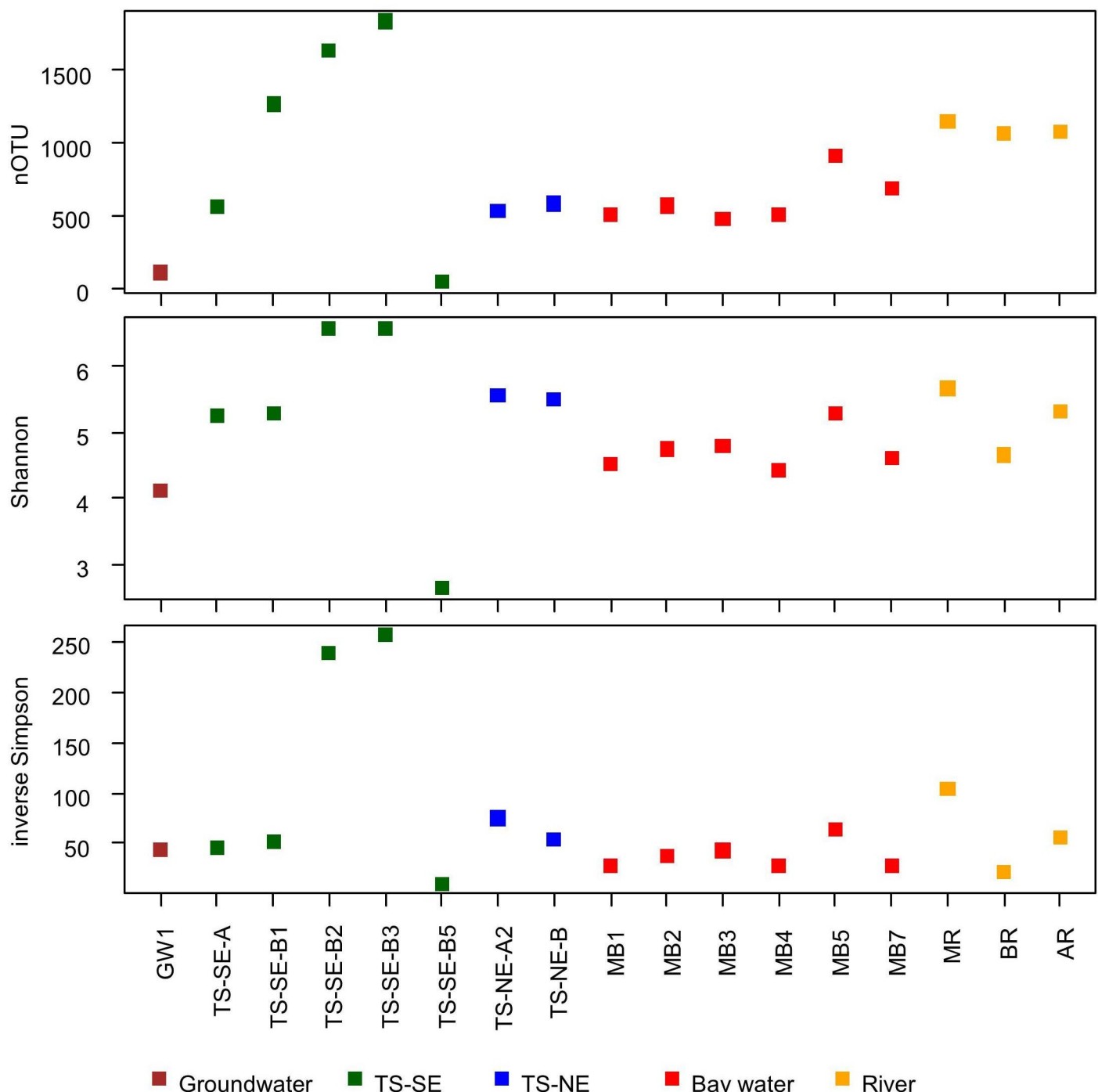

**Fig 3.** Alpha diversity of microbial communities across all samples assessed by number of OTUs (A), Shannon (B) and inverse Simpson indices (C). Samples TS-SE-B4, TS-NE-B1, and MB6 were not included in the calculation due to insufficient sequencing depth.

TS-SE-B3 consistently had the highest number of OTUs, Shannon, and inverse Simpson indices compared to the other samples, while the lowest number of OTUs, Shannon and inverse Simpson indices was found in TS-SE-B5. This pattern was not visible in GW-1, which exhibited the second lowest number of OTUs, but average Shannon and inverse Simpson indices. The remaining subsurface samples, as well as bay and river water samples were within a similar richness and diversity range. The Spearman correlation analysis between subsurface taxonomic diversity and observed environmental parameters indicated that DON had the highest correlation to Shannon and inverse Simpson indices (Spearman ρ = -0.62), followed by $PO_4^{3-}$ concentration (Spearman ρ = 0.52) (S1 Table).

Bray-Curtis (BC) dissimilarities between samples ranged from 0.2 to 0.9 (Fig 4). Generally, each surface water sample from the river and the Mobile Bay water column was quite similar to each other in terms of microbial community composition (pairwise BC = 0.3–0.5). Microbial community composition in MB5-7 was more similar to river samples than MB1-4. All samples from the vertical SGD profiles were highly heterogeneous (pairwise BC > 0.6). Across the TS-SE samples, microbial communities among TS-SE-B1 and TS-SE-B2 were more similar to each other, while the deeper samples (TS-SE-B3, TS-SE-B4, TS-SE-A) displayed a higher heterogeneity. Within the TS-NE area, microbial communities of TS-NE-A2 and TS-NE-B were more similar to each other than to TS-NE-A1 due to similar depth. Overall, the average dissimilarity within subsurface samples (BC = 0.9) was higher than within surface samples (BC = 0.8).

Fitting environmental parameters onto the NMDS ordination for surface water samples showed that microbial community composition in river and northern Mobile Bay (MB5-7)

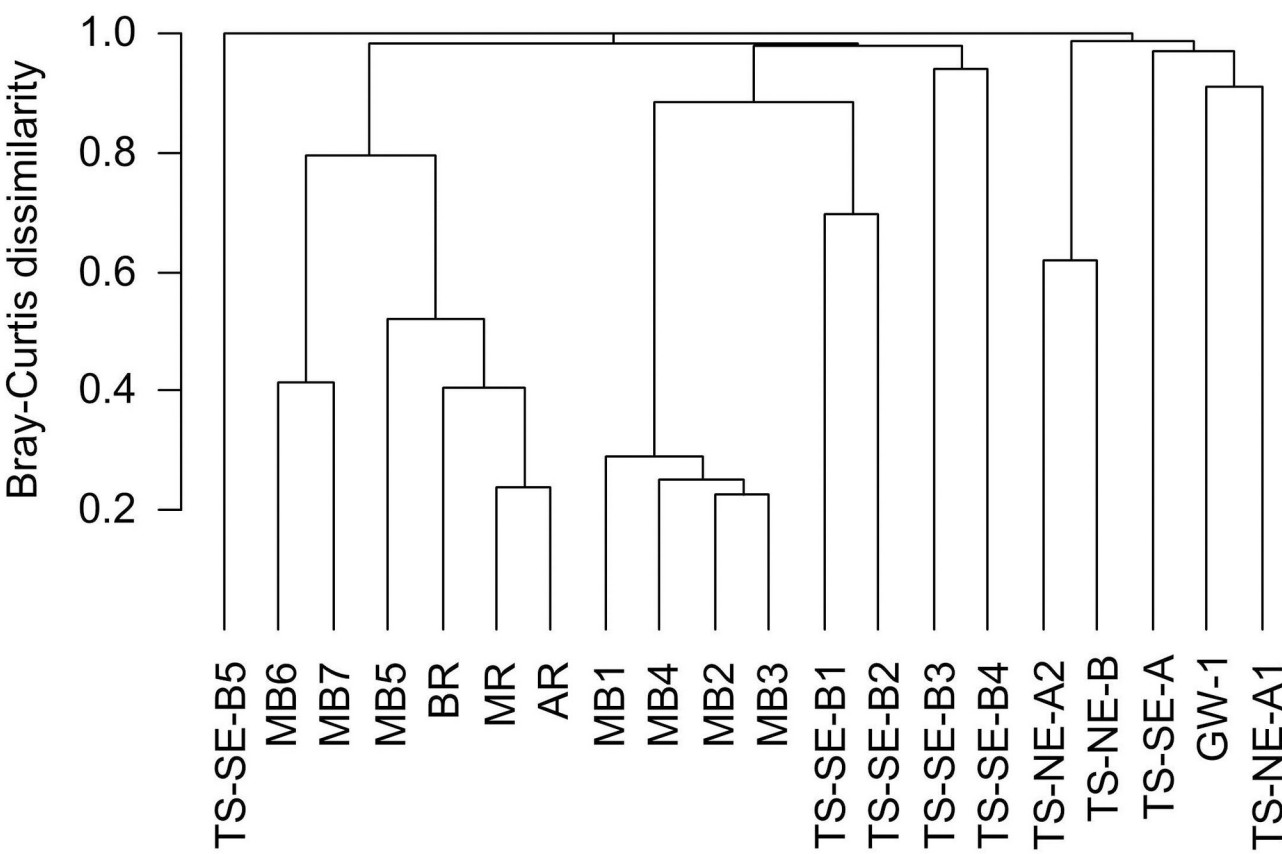

**Fig 4. Complete linkage hierarchical clustering based on Bray-Curtis dissimilarities of relative OTU proportions.**

water samples were associated with high nutrient concentration (Fig 5A), while the communities in the southern Mobile Bay (MB1-4) were positively associated with salinity. The patterns in microbial community composition of the subsurface samples and their association with environmental parameters were explored in separate NMDS ordination (Fig 5B), which showed that TS-NE microbial samples were positively associated with DO and $NO_3^-$. The majority of TS-SE microbial samples were associated with $NH_4^+$, $PO_4^{3-}$, DON, salinity, and temperature, except for TS-SE-B5. TS-SE-B5 and GW1 were located the furthest from the other samples in the ordination, indicating their distinct microbial community composition.

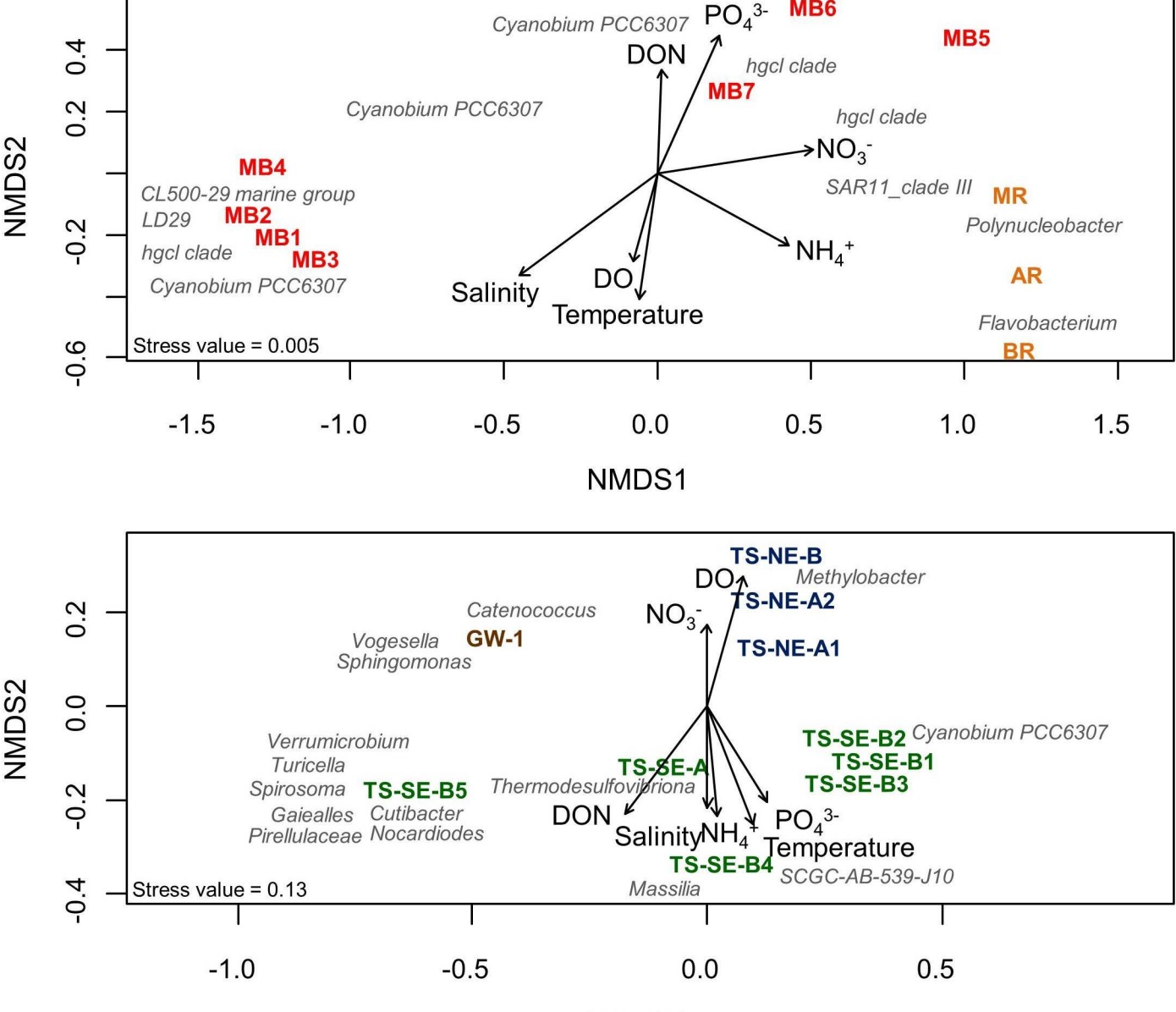

**Fig 5.** Non-metric multidimensional scaling (NMDS) plot depicting the association between microbial community composition and environmental data for surface water samples (A) and subsurface water samples (B). The position and taxonomic affiliation of dominant OTUs most fitted to the observed environmental parameters are shown.

Salinity and $NH_4^+$ were identified as the strongest predictors of community shifts in surface samples, in total explaining 58.5% of the variation in community composition (Table 2). Salinity was determined as most significant parameter with a pure contribution of 26.9% to the change of microbial community composition across all surface samples. Due to some collinearity, there was a 17% overlap in explained variation between these two parameters. Among the subsurface samples, DON and $PO_4^{3-}$ were determined as best-suited parameters to explaining patterns in microbial community composition with a total contribution of 25.4%. Both parameters had equal pure contributions, and there was no overlap in explained variation. The selection of DON and $PO_4^{3-}$ as significant predictors implied a potentially similarly important role of DO and salinity, as any effect attributed to DON and $PO_4^{3-}$ could also be explained by DO and salinity due to their high correlation (DON and DO: Pearson r = -0.74, $PO_4^{3-}$ and salinity: Pearson r = 0.98).

## Microbial community composition across horizontal and vertical scales

We found that *Gammaproteobacteria*, *Bacteroidia*, *Oxyphotobacteria*, *Actinobacteria*, and *Alphaproteobacteria* were the most dominant bacterial classes across all samples (Fig 6A). *Gammaproteobacteria* were present in high proportions in the inland groundwater (70%), river water, and TS-NE samples (average 20%). OTUs affiliated with the freshwater genera *Vogesella* and *Polynucleobacter* of the *Gammaproteobacteria* occurred in high proportions in the inland groundwater sample. OTUs related to *Methylobacter* (class *Gammaproteobacteria*) and unclassified *Thermodesulfovibrionia* were the most dominant OTUs from TS-NE and TS-SE, respectively. From subsurface to surface water samples, community composition shifted from *Gammaproteobacteria*-dominated to a community with increased proportions of *Oxyphotobacteria*, *Bacteroidia*, *ActinobacteriaI* and *Alphaproteobacteria*. *Oxyphotobacteria* dominated bay water samples and the upper two layers of the TS-SE vertical profile (TS-SE-B1, TS-SE-B2). The most dominant OTU of the *Oxyphotobacteria* was classified as *Cyanobium PCC-6307*. Within the surface samples, this OTU was identified as the most dominant and best fitted to salinity, $NO_3^-$, $NH_4^+$, and $PO_4^{3-}$ (Fig 5A). *Bacteroidia* were mainly represented by a *Flavobacterium* OTU with the highest proportions in river samples (Fig 6B). OTUs from *Actinobacteria* were almost exclusively detected in surface water samples, mostly affiliated with the *hgcl clade*. *Alphaproteobacteria* were detected in high proportions across surface water samples with an average of 15% and 8% in river and coastal samples, respectively. OTUs of the *SAR11 clade* comprised 50% of the *Alphaproteobacteria* in surface water samples with *SAR11 clade II* predominantly found in MB1-4 and *SAR11 clade III* occurring mainly in MB5-7 and river samples, respectively.

**Table 2. Contribution and significance of observed environmental factors shaping microbial community composition based on PERMANOVA.**

| Source of variation | Adjusted $R^2$ (%) | Df | F | P-value |
|---|---|---|---|---|
| Surface samples | | | | |
| Final model | 58.5 | 2 | 4.92 | 0.001*** |
| Salinity | 26.9 | 1 | 4.54 | 0.004** |
| $NH_4^+$ | 12.9 | 1 | 2.18 | 0.072. |
| Subsurface samples | | | | |
| Final model | 25.4 | 2 | 1.19 | 0.022* |
| DON | 12.9 | 1 | 1.21 | 0.044* |
| $PO_4^{3-}$ | 12.6 | 1 | 1.18 | 0.063. |

Significance codes: 0 '***' 0.001 '**' 0.01 '*' 0.1 '.' 0.1

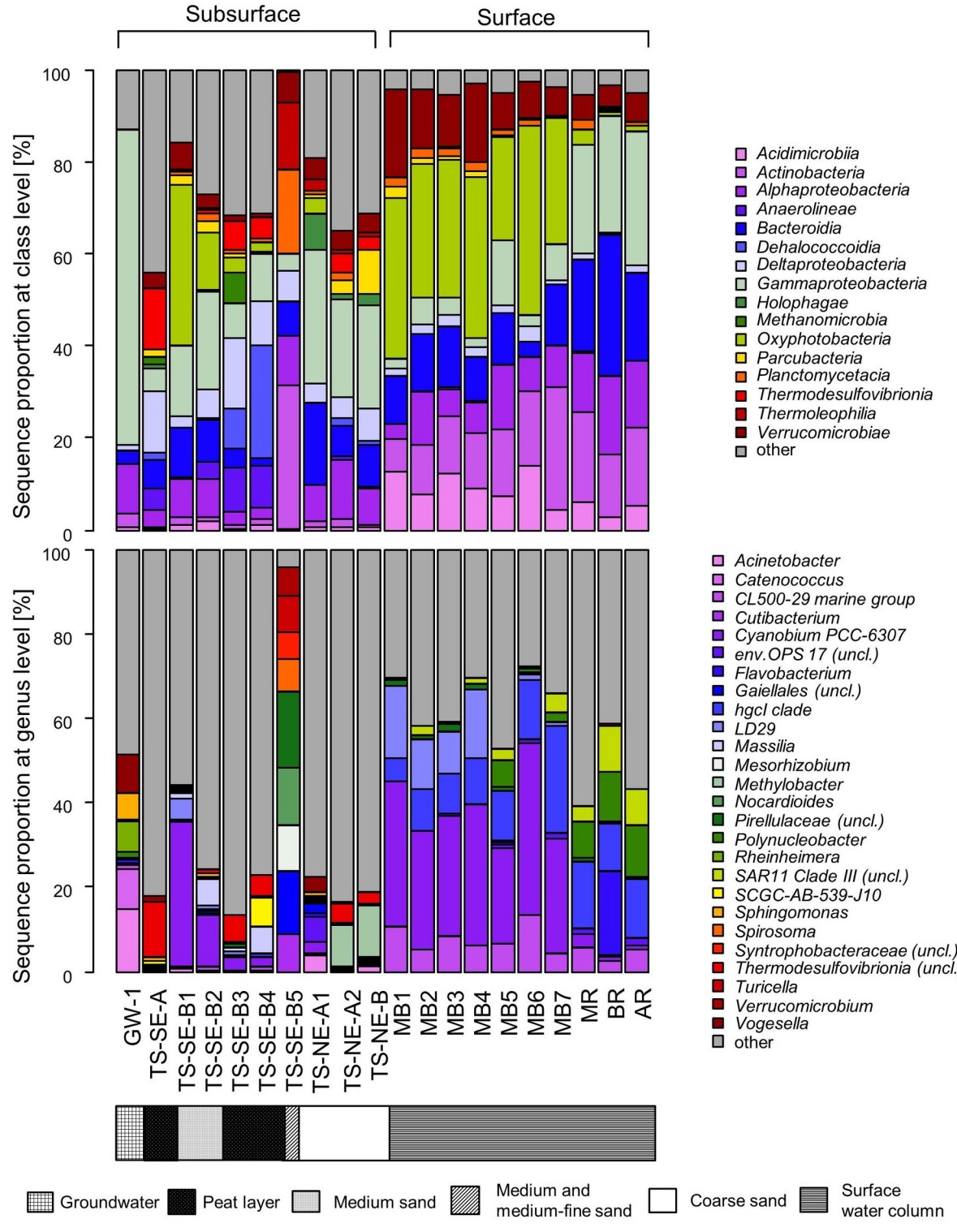

**Fig 6.** Relative sequence proportion of dominant microbial taxa at class (A) and genus level (B).

At TS-SE, the microbial community composition differed notably between the upper (TS-SE-B1 and TS-SE-B2), middle (TS-SE-B3, TS-SE-B4), and bottom sediment layers

(TS-SE-B5). *Cyanobium PCC-6307*, which displayed high proportions in all bay water samples, dominated TS-SE-B1 (9%) and TS-SE-B2 (17%). The TS-SE-B2 sample originated from an intermediate layer between the oxic upper layer and anaerobic deeper layers, and was characterized by e.g. aerobic cyanobacteria and anaerobic sulfate-reducing bacteria (SRB). The middle layers in TS-SE were dominated by a mixture of obligate anaerobic bacteria, fermentative bacteria, SRB, and methanogenic archaea, indicative of a highly reducing environment. Unclassified *Thermodesulfovibrionia* showed the highest proportion in TS-SE-A (13%), and also notably contributed to TS-SE-B3 and TS-SE-B4 (5–6%). This taxon also had the highest *ordiselect* score according to the ordination of the subsurface samples, indicating their dominance and association with changes in environmental parameters, particularly to DON, DO, and temperature. Another dominant genus comprising 2 to 25% in the peat layer was SCGC-AB-539-J10 of the *Dehaloccoccoidia*, which is one of the most widely distributed taxa in marine subsurface sediments samples. We also identified high proportions of SRB-related genera of the class *Deltaproteobacteria*: unclassified *Syntrophobacteraceae*, *Desulfobacca*, *Desulfatiglans*, *Syntrophorhabdus*, *Syntrophus*, and unclassified *Desulfobulbaceae*. Overall, the aforementioned genera accounted for 12%, 9%, 4% of the sequences in TS-SE-B3, TS-SE-B4, and TS-SE-B5 samples, respectively. Unclassified *Anaerolineaceae* of the class *Anaerolinea*, a taxon consisting of obligate anaerobic and fermentative bacteria, contributed between 1 and 9% in these layers. The dominant taxa in the deepest sediment layer (TS-SE-B5), were found to be distinctly different from the upper layers and comprised mainly of the following genera: unclassified *Pirellulaceae* (18%), unclassified *Gaiellales* (15%), *Nocardiodes* (14%), and *Mesorizhobium* (11%).

At the TS-NE site, *Gammaproteobacteria* was the most dominant bacterial class in all layers, followed by *Deltaproteobacteria* and *Alphaproteobacteria*. At genus level, most of the *Gammaproteobacteria* in these samples were related to type I methanotrophs of the genera *Methylobacter* and *Methylococcus*. In total, methanotrophic genera accounted for 12% and 15% of the sequences in TS-NE-A2 and TS-NE-B, respectively. Overall, the TS-NE vertical profile shifted from a mixture of marine (e.g. unclassified *env. OPS17* and *Cyanobium PCC-6307*) and facultative anaerobic bacteria (e.g. *Geothrix*) in the uppermost sediment layer to a more methanotroph-dominant community in the bottom layer.

Across all samples, archaea were identified in notable proportions at the anaerobic TS-SE site, i.e. 4%, 4%, 11%, and 6% in TS-SE-A, TS-SE-B3, TS-SE-B4, and TS-SE-B5, respectively. *Methanomicrobia*, *Omnitrophicaeota*, and *Woesearchaeia* were the dominant archaeal classes found in the samples. *Methanomicrobia* were mainly represented by the genera *Methanolinea*, *Methanoregula* and *Candidatus* Methanoperedens. Unclassified *Bathyarchaeia* were found in TS-SE-B3 and TS-SE-B4 together with *Methanomicrobia*. To a lesser extent, archaea were also found in TS-SE-A1, TS-NE-A2, and TS-NE-B with a sequence proportion of 1–2%. At these sites, *Thaumarchaeota* was the dominant class, mainly represented by nitrifying genera *Candidatus* Nitrosopumilus and unclassified *Nitrosotaleaceae*.

## Metabolic prediction

To accompany the result of the physico-chemical and microbial community analysis, we used Tax4Fun2 to predict microbial metabolic functions in the subterranean estuary that may affect the composition of SGD. Tax4Fun2 was able to use on average 54% of the sequences represented in 44% of the OTUs for metabolic prediction. The highest unused fraction for the metabolic prediction was found in TS-SE-A. The highest proportions of KOs related to ammonia oxidation (nitrification) were predicted in TS-NE-A2 and TS-NE-B and the lowest in TS-SE-B5 and MB1-4 (Fig 7). Communities of inland groundwater and the deeper layers of

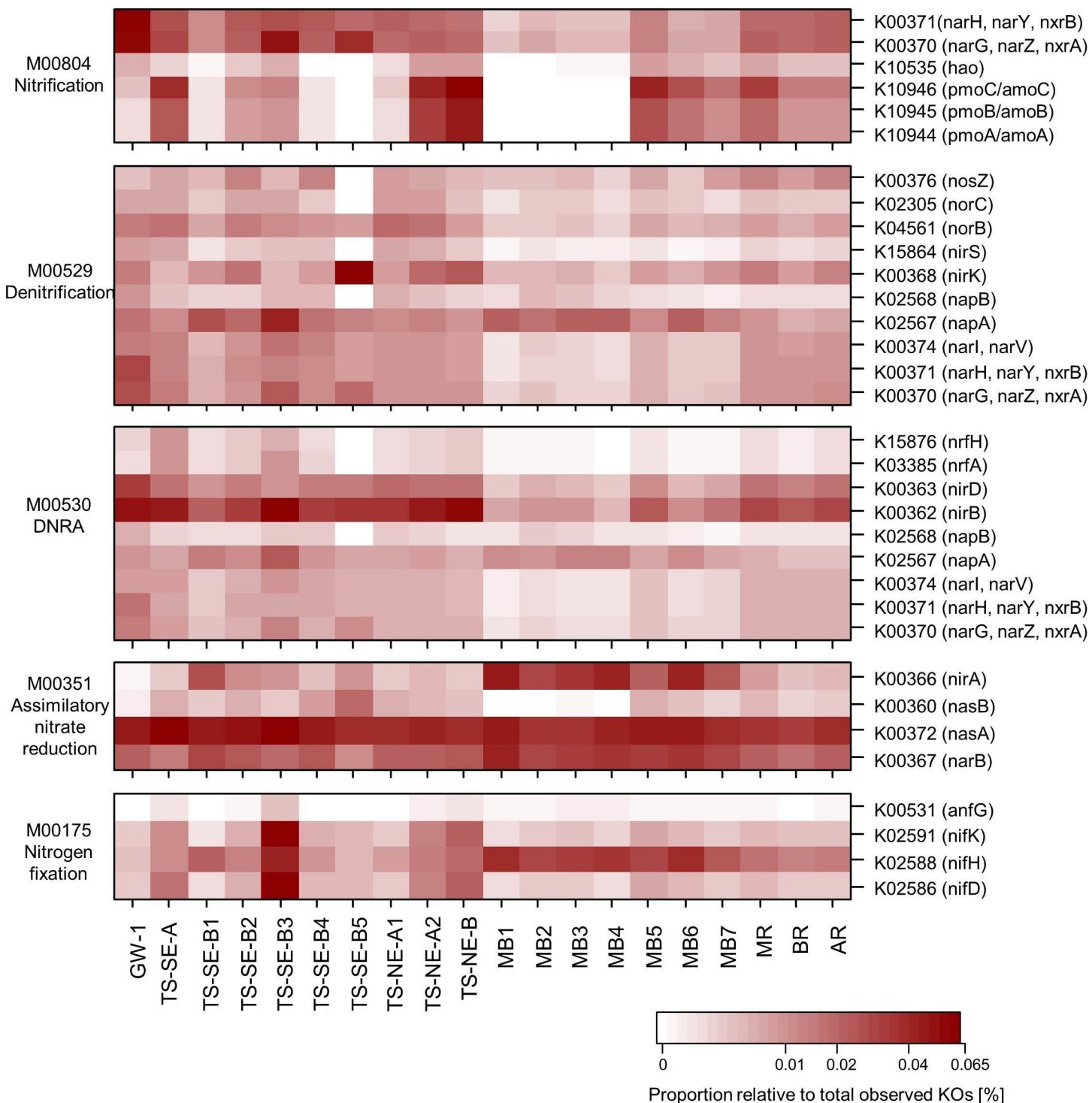

**Fig 7. Heatmap of the proportion of predicted KEGG Orthologs (KO) related to nitrogen metabolism based on Tax4Fun2.** Dissimilatory nitrate reduction to ammonium: DNRA.

TS-SE were characterized by nitrogen cycling pathways usually occurring in anoxic environments, such as denitrification, DNRA, or nitrogen fixation. Except for assimilatory nitrate

reduction, enzymes related to the nitrogen cycle were predicted in lower proportions in surface waters compared to subsurface samples.

## Discussion

### Changes in microbial community composition along the freshwater-marine water continuum

In this study, we observed significant variations in microbial community composition and diversity across a horizontal (e.g. groundwater-river-bay) and vertical (e.g. pore water depth profile) gradient in the STE of Mobile Bay, Alabama. Among the observed environmental parameters, salinity was identified as the primary factor affecting the variation of microbial community composition in the surface water samples. The importance of salinity in governing the distribution and diversity of microbial communities across the hydrological continuum is well studied [33, 35, 54]. Across the subsurface samples, DON and $PO_4^{3-}$ were identified as the parameters explaining the observed patterns in community composition with similar amount of contribution. DON is normally generated in TS-SE by peat mineralization where its concentration was considerably higher in comparison to TS-NE, and likely resulted in the different community composition between the two locations. The influence of DON in shaping microbial community composition is also found in other study conducted in peatlands [55], while $PO_4^{3-}$ is also identified as a limiting nutrient for subsurface microbial communities in peatlands as inorganic orthophosphate tends to bond to organic compound; therefore restricting its bioavailability [56]. Furthermore, in this case DON and $PO_4^{3-}$ may act as a proxy for DO and salinity due to their high correlation in subsurface samples. DO as a potential predictor of microbial community composition in subsurface samples is expected due to the steep gradient from aerobic to anaerobic conditions across these samples. Indeed, all of the TS-NE samples were considered more oxygenated due to the higher hydraulic conductivity and coarser lithology than TS-SE. DON and $PO_4^{3-}$ concentration affecting the diversity of the subsurface microbial communities was also supported by correlation analysis between alpha diversity and environmental parameters. Nevertheless, as only 25% of community variation could be attributed to DON and $PO_4^{3-}$, it is assumed that other environmental variables play additional roles in this environment. Considering that we found high proportions of SRB, methanogenic and methanotrophic bacteria and archaea, it is likely that the concentrations of various electron acceptors strongly determine subsurface communities [34]. For example, in a subterranean environment inhabited by both SRB and methanogenic archaea, their abundance is controlled by available substrate concentration (e.g. acetate, hydrogen, methanol), organic loading [57, 58], and ion concentration [59], which were not measured in this study. In this case, further studies, including measuring more environmental parameters, are needed to characterize the drivers of community shifts with higher certainty.

In general, the overall proportion of explained variation in the subsurface samples was lower compared to surface samples due to the high heterogeneity of subsurface microbial communities. Pore water samples from the TS-SE consisted of a diverse mixture of fresh water and marine microbes, as well as aerobic, facultative anaerobic, and obligate anaerobic bacteria, thus increasing microbial alpha diversity, especially in TS-SE-B2 and TS-SE-B3. The higher diversity of the communities in the anaerobic layers is also supported by the higher proportion of archaea identified in these layers, which grow by fermentation or anaerobic respiration using various electron acceptors. TS-SE-B1 exhibited a high number of OTUs but average Shannon and inverse Simpson indices, indicating the presence of a large number of rare OTUs in this sample. Inversely, GW-1 had average Shannon and inverse Simpson indices but low number of OTUs, pointing to a more even community. The low diversity of OTUs in GW-1

may also arise from limited substrate availability (e.g. organic matter), lack of photosynthesis, or as a result of filtration processes through the aquifer material [60, 61].

In this study, the most notable difference between the inland groundwater microbial community and the rest of the samples is the high proportion of *Gammaproteobacteria*, particularly from the families *Chromobacteriaceae*, *Burkholderiaceae*, and *Pseudomonadaceae*, which are all common in fresh groundwater samples [62, 63]. The genus *Pseudomonas*, from the family *Pseudomonadaceae*, is known for its denitrifying capabilities in almost all of its species [64]. The occurrence of denitrification in the aquifer is also supported by metabolic prediction. Such a microbial contribution may be relevant considering that Montiel, Lamore (24) reported denitrification as one major pathway of $NO_3^-$ removal in the coastal aquifer of Mobile Bay.

We observed a phylogenetic shift from subsurface to surface water communities, the latter resembling eutrophic planktonic communities presumably due to high primary productivity and frequent contact with anthropogenic activities and contamination [63]. The microbial communities in surface water in this study were dominated by cyanobacteria and other bacterioplankton taxa. In river samples, microbial community composition was characterized by a high proportion of *Actinobacteria* and *Bacteroidia*, which are common lineages in surface freshwater [65, 66]. In this study, they were represented by the *hgcl clade* and *Flavobacterium*, respectively. The *hgcl clade* is a ubiquitous taxon and associated with N-rich environments or organic compound utilization [67], while *Flavobacterium* is widely known to play an essential role in the degradation of complex biopolymers in marine and freshwater environment [68]. The most dominant surface water genus from *Gammaproteobaceria*, *Polynucleobacter*, is often found in surface water bodies impacted by urban activities [69, 70].

The microbial community composition of the river samples is indicative of a transition zone between fresh and saline surface water as well as an active tidal cycle. There is a combination of brackish-marine taxa (e.g. *CL500-29 marine group*, *SAR11 clade III*) with freshwater taxa that were not found in the more saline samples (e.g. *Limnohabitans*, *Sediminibacterium*, *Pseudocarcicella*). We found that the extent of the river plume into Mobile Bay was traceable by both salinity and microbial community composition. During the sampling campaign in July 2017, the salinity increased from 0.2 in the northern part of the bay and increased into 4.7 in the southern part. Furthermore, beta diversity analysis suggested that the microbial communities of the tributary rivers (BR, MR, AR) were similar to those from the northern part of Mobile Bay (i.e., MB5-7), indicating that the northern part of the bay was heavily influenced by river discharge which was also reported by Ortmann and Ortell [42].

Water samples from Mobile Bay were characterized by OTUs related to the *SAR11 clade*, *Cyanobium PCC6307*, and *CL500-29 marine group*. *Cyanobium PCC-6307* is a freshwater cyanobacterium commonly found in warm water [71]. The distribution of *SAR11 clade* was distinctly associated with their salinity preference: while *SAR11 clade II* was dominant in the southern part of Mobile Bay and the clade itself is commonly associated with coastal water, the northern part of Mobile Bay had a higher proportion of *SAR11 clade III* known from brackish water communities [72]. The dominance of SAR11 in the lower end of estuaries and low salinity coastal water was also reported from other studies from Mobile Bay [42, 73, 74]. All of the above bacterial taxa are associated with active primary productivity and carbon cycling [67, 72, 75]. *CL500-29* is also known from algal blooms and eutrophic waters [76–78]. In an environment of high primary productivity, cyanobacteria and heterotrophic bacteria may be involved in close interactions. The cyanobacteria release organic matter that attracts heterotrophic bacteria and inversely the heterotrophic bacteria may release inorganic nutrients which could be recycled and reused by cyanobacteria [79, 80]. The occurrence of high primary productivity suggested by microbial community composition in this study, in addition to vertical stratification of the water column due to natural wind and tide pattern of Mobile Bay [39, 81] and

elevated SGD-derived DON and $NH_4^+$ [24], may promote the occurrence of hypoxia events during the dry seasons.

In general, we did not observe any occurrence of community coalescence between freshwater and marine bacteria except for the prevalence of marine taxa in river samples and the uppermost layer of the pore water vertical profiles. These marine taxa are most likely brought to the estuaries and the upper STE by the previous high tide, and their high proportion in a zero salinity environment suggests a tolerance to a wide range of salinities [82]. Inversely, freshwater bacteria are found only in low or even negligible proportions in the brackish and saline samples. In the bay water sample collected close to groundwater discharge points (i.e. MB6, which is located in the direction of flow discharge from TS-NE), we did not identify freshwater taxa or taxa that are dominant in the TS-NE pore water. This finding contradicts a study from Lee, Shin [36], which found subsurface bacteria in coastal water during ebb tide characterized by high SGD rate. The different findings could be caused by the location of our sampling point, which is too far from the land that freshwater taxa exported by SGD or surface water might already be lost due to environmental and biotic filtering, considering that these taxa have a lower resistance to salinity changes in comparison to marine taxa [82]. Nevertheless, this highlights the importance of implementing microbial studies over a more extended time period (i.e. covering different tidal cycle or seasons).

## Vertical distribution of microbial community composition in SGD sites

To describe vertical patterns in microbial community composition, we sampled vertical pore water profiles from two SGD sites. A sediment core recovered from the TS-SE, that essentially reflects the STE in this site, showed a lithological stratification. The uppermost layer consists of medium sand, followed by organic-rich/peat layer, and medium sand at the bottom of the core. The upper layers were dominated by aerobic bacteria and marine cyanobacteria, which suggests the infiltration of surface bay water into the upper part of the intertidal zone. The peat layer had distinctive anaerobic environment with a high proportion of SRB, fermentative bacteria, and methanogenic archaea, indicating the prevalence of syntrophic hydrocarbon degradation. In a reducing environment, it is common to find syntrophic consortia of these three types of bacteria [34, 57, 83–85]. Organic matter from the peat layer can be hydrolyzed and fermented by fermentative bacteria, whose end products serve as the primary substrates for sulfate reduction or methanogenesis [86]. Even though the SRB and methanogens compete with each other to utilize the organic substrates produced by the fermentative bacteria, their coexistence in peatlands is not uncommon [58, 87, 88]. The three most dominant archaeal classes in these reducing layers were *Woesearchaeia*, *Bathyarchaeia*, and *Methanomicrobia*. *Woesearchaeia*, commonly found in groundwater and sediment, are known for their symbiotic lifestyle and role in anaerobic carbon cycling [31]. The predominant genera from class *Methanomicrobia*, *Methanolinea* and *Methanoregula*, are commonly found in other reducing environments such as peat bogs or anaerobic wastewater sludge [89, 90]. The co-occurrence of *Methanomicrobia* with *Bathyarchaeia* in the anoxic layers is also reported from peatlands in China [85] as well as from engineered anaerobic sludge systems [91]. In peatlands, the symbiotic association of *Methanomicrobia* and *Bathyarchaeia* plays a critical ecological role in the production of acetate, an essential precursor of methanogenesis. The potential occurrence of the productive methane production in the reduced layer of TS-SE is supported by the result of methane-related metabolic prediction (S1 Fig). While methane concentration was not measured in this study, a previous study found methane fluxes were detected in some parts of the Mobile Bay coastline, with stronger fluxes identified in lower salinity coastal areas [92].

The bottom-most layer of the TS-SE vertical profile was characterized by typical soil or sediment bacteria (e.g. *Mesorizhobium*, Delgado, Casella [93]). The distinct change in microbial community composition as well as the predicted metabolic function suggests that the bottom sand layer is independent of the above peat layer.

At the TS-SE site, $NO_3^-$ rich groundwater percolates through an organic-rich/peat sediment layer and the bulk nitrogen is transformed into $NH_4^+$ before discharging to Mobile Bay. In addition to net $NH_4^+$ production by mineralization, the loss of $NO_3^-$ in the peat layer may be related to biological uptake (assimilatory nitrate reduction), denitrification or DNRA, as supported by the metabolic prediction. $NO_3^-$ losses through denitrification in peat soils have been observed elsewhere (e.g. Van Beek, Hummelink [94], Jörgensen and Richter [95]). Denitrification is regulated by the *nirS/nirK* subunits of the nitrite reductase gene in bacteria, which is widespread among various taxonomic groups of bacteria except for gram-positive bacteria and obligate anaerobes, with few exceptions [96, 97]. DNRA is mostly carried out by fermentative or chemolithotrophic bacteria recycling $NO_3^-$ to $NH_4^+$ facilitated by the genes encoded as *nrfA*, *nirD* and *nirB* [98, 99]. These three genes work complementarily in reducing nitrate to ammonia depending on the condition of low/high nitrate input [100]. The occurrence of DNRA in Mobile Bay's STE has been reported before in the adjacent smaller estuary Weeks Bay [30]. The prevalent taxa found in this site, i.e. *Dehaloccoidia*, *Bathyarchaeia*, *Thermodesulfovibrionia*, *Anaerolinaceae*, *Syntrophaceae*, and *Syntrophorhabdus* were reported to be associated with DNRA processes in other studies conducted in sediments with similar characteristics [101–104]. *Deltaproteobacteria*-affiliated SRB, whose proportion reached up to 12% in the peat layer, have the secondary capacity to implement DNRA due to their ability to grow with nitrate as a terminal electron acceptor and reduce it to ammonium [105–107]. In many cases, the coexistence of denitrification and DNRA processes is found in oxic-anoxic coastal interfaces [102, 108], where oxidation status and C/N ratio often determine which process dominates N cycling [96, 109, 110]. This finding also supports other microbial studies stating that STEs are a biogeochemical hotspot for nitrogen cycling and these complex processes may alter SGD quality discharging to coastal waters (e.g. Santoro, Francis [28], Adyasari, Hassenrück [35]).

At the TS-NE site, the lithology consists exclusively of coarse sand and is distinguished by a high hydraulic conductivity. The microbial community at this site consisted of a mixture of aerobic and facultative bacteria, with a high proportion of methanotrophic bacteria of the genus *Methylobacter*. The *Methylobacter* lineage consists of some of the most abundant methane oxidizers in soil communities, which commonly utilize methane or other C1-compounds [111, 112]. This was in accordance with the organic source analysis from the peat layer that was close to, but not in contact with, the piezometer, and composed of C1 and C2 compounds indicative of decaying plant material [24]. Methanotrophs usually contribute to biogeochemical cycling by oxidizing methane or ammonia using the enzymes methane monooxygenase (encoded by the gene *mmo*) and ammonia monooxygenase (*amo*), respectively. Both *amo* and *mmo* have similar homology and may interchangeably be used by the methanotrophs [112–115]. Notably, the proportion of predicted ammonia oxidation pathways in TS-NE-A2 and TS-NE-B was among the highest across all of the samples.

Overall, the total sequence proportion of archaea across all of the samples only represents 1% of the microbial community. Archaea generally compose a minority fraction of the microbial community in groundwater, however they are known to play a notable role in shaping the biogeochemical cycle in their respective environment regardless of their low relative abundance [28, 116], which is supported by the result of metabolic prediction. For instance, the archaeal community seems to contribute to ammonia oxidation at TS-NE, indicated by the prevalence of *Thaumarchaeota*, a ubiquitously distributed archaeal class known for their

ammonia-oxidizing capability [32, 117]. Furthermore, the methane cycling processes at TS-SE may likely be regulated by the methanogenic archaea. This finding is also consistent with other studies reporting rare taxa to contribute disproportionately and perform essential biogeochemical functions in the ecosystem [118, 119].

As a limitation of the metabolic prediction, it should be noted that the usage of short 16S rRNA gene sequences to predict metabolic functions is severely constraint by the limited information content of the amplicon, biasing the accurate assignment of functional genes to a specific 16S. Thus, we suggest using the results of this study as a primer for further metagenomic or metatranscriptomic investigations, as well as quantitative polymerase chain reaction (qPCR) assays targeting, e.g., methanotrophic, methanogenic bacteria and archaea, which displayed notable sequence proportion in the coastal pore water samples. Methane is highly abundant in peatlands [120, 121] and one of the main contributors of aquatic greenhouse gases emissions [122, 123]. Thus, understanding its cycling may be valuable particularly in coastal areas with dynamic sea-level change and a fast-changing anthropogenic landscape such as Mobile Bay.

## Conclusion

In this study, we assessed microbial community composition and distribution in terrestrial and coastal water of Mobile Bay, Alabama, where SGD-derived nutrient fluxes cause harmful algal blooms and large scale fish and crustacean kills (*Jubilee* events) each summer. Microbial community composition across the surface hydrological continuum was primarily governed by salinity, while DON and $PO_4^{3-}$ concentrations were the strongest predictors of community shift within the subsurface ecosystems. Samples originating from the estuary and water column of Mobile Bay were dominated by bacterioplankton, particularly cyanobacteria, which are typically found in environments with high primary productivity. We further observed that microbial communities displayed different compositions along the two vertical profiles at SGD sites due to contrasting sediment stratigraphy and presumed oxygen distribution. This study suggest that microbial communities in these two sites are essential contributors to the nitrogen and methane cycle, particularly related to denitrification and DNRA at the TS-SE site, and ammonia or methane oxidation at the TS-NE site. Considering that the microbially mediated elemental cycling in the STE influences the chemical composition of the submarine groundwater discharging to Mobile Bay's water column, these findings may be used as an incentive for further studies involving a wide range of metabolic processes and their potential impact on the ecosystem in terms of climate-related hydrological change on a regional scale.

## Supporting information

**S1 Fig. Heatmap of the proportion of predicted KEGG orthologs (KO) related to methane metabolism based on Tax4Fun2.** M00356: methanol to methane; M00357: acetate to methane; M00563: methylamine/dimethylamine/trimethylamine to methane; M00567: $CO_2$ to methane. (TIF)

**S1 Table. Spearman correlation between alpha diversity and observed environmental parameters.**
(DOCX)

## Acknowledgments

We thank Yustian Alfiansyah for his help with microbial laboratory analysis. We also acknowledge Weeks Bay National Estuarine Research Reserve and the Army Corp of Engineers in Mobile for providing technical support during the field campaign.

## Author Contributions

**Conceptualization:** Dini Adyasari, Daniel Montiel, Natasha Dimova.

**Data curation:** Dini Adyasari.

**Formal analysis:** Dini Adyasari, Christiane Hassenrück, Daniel Montiel.

**Funding acquisition:** Natasha Dimova.

**Investigation:** Dini Adyasari, Daniel Montiel, Natasha Dimova.

**Methodology:** Dini Adyasari, Christiane Hassenrück.

**Project administration:** Natasha Dimova.

**Resources:** Natasha Dimova.

**Supervision:** Natasha Dimova.

**Validation:** Christiane Hassenrück.

**Visualization:** Dini Adyasari, Christiane Hassenrück, Daniel Montiel.

**Writing – original draft:** Dini Adyasari.

**Writing – review & editing:** Dini Adyasari, Christiane Hassenrück, Daniel Montiel, Natasha Dimova.

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
