## [Decision Letter · Decision Letter 0]

14 Apr 2020

PONE-D-20-05547

Microbial community composition across a coastal hydrological system affected by submarine groundwater discharge (SGD)

PLOS ONE

Dear Dr. Adyasari,

Thank you for submitting your manuscript to PLOS ONE. After careful consideration, we feel that it has merit but does not fully meet PLOS ONE’s publication criteria as it currently stands. Therefore, we invite you to submit a revised version of the manuscript that addresses the points raised during the review process.

Two reviewers have provided excellent reviews. They have both concluded that the content of the paper is worthy of publication, but have suggested that the results could be explored more deeply (see comments from Reviewer 2) work should be presented more clearly and in a more organized fashion (see comments from Reviewer 1).  They make several suggestions that should help you achieve those goals.

We would appreciate receiving your revised manuscript by May 29 2020 11:59PM. To enhance the reproducibility of your results, we recommend that if applicable you deposit your laboratory protocols in protocols.io, where a protocol can be assigned its own identifier (DOI) such that it can be cited independently in the future. For instructions see: http://journals.plos.org/plosone/s/submission-guidelines#loc-laboratory-protocols

We look forward to receiving your revised manuscript.

Kind regards,

John M. Senko

Academic Editor

PLOS ONE

2. Please amend your Competing interests statement to declare any competing interests, such as author commercial affiliations.

3. In your Methods section, please provide additional location information, including geographic coordinates for the data set if available.

4. In your Methods section, please provide additional information regarding the permits you obtained for the work. Please ensure you have included the full name of the authority that approved the field site access and, if no permits were required, a brief statement explaining why.

6. Thank you for stating the following in the Competing Interests section:

We note that one or more of the authors are employed by a commercial company: Geosyntec Consultants

7.  We note that Figure 2 in your submission contains map images which may be copyrighted. All PLOS content is published under the Creative Commons Attribution License (CC BY 4.0), which means that the manuscript, images, and Supporting Information files will be freely available online, and any third party is permitted to access, download, copy, distribute, and use these materials in any way, even commercially, with proper attribution. For these reasons, we cannot publish previously copyrighted maps or satellite images created using proprietary data, such as Google software (Google Maps, Street View, and Earth). For more information, see our copyright guidelines: http://journals.plos.org/plosone/s/licenses-and-copyright.

1.    You may seek permission from the original copyright holder of Figure 2 to publish the content specifically under the CC BY 4.0 license. 

Reviewers' comments:

Reviewer's Responses to Questions

**Comments to the Author**

1. Is the manuscript technically sound, and do the data support the conclusions?

Reviewer #1: Partly

Reviewer #2: Yes

2. Has the statistical analysis been performed appropriately and rigorously? 

Reviewer #1: Yes

Reviewer #2: Yes

3. Have the authors made all data underlying the findings in their manuscript fully available?

Reviewer #1: Yes

Reviewer #2: Yes

4. Is the manuscript presented in an intelligible fashion and written in standard English?

Reviewer #1: Yes

Reviewer #2: Yes

5. Review Comments to the Author

Reviewer #1: SGD has been recognized as a major pathway of inland-derived nutrients and chemical flux into ocean. Nitrogen transformation mediated by chemical and microbial reaction along its pathway from inland aquifer to subterranean estuary could serve as a major mechanism controlling the level and form of nutrient discharge into the ocean. This paper aims to investigate microbial composition and distribution along the coastal subsurface/surface water bodies influenced by nutrient-rich SGD. The manuscript presents interesting results (sharp shifts in microbial composition depending on location and environmental factors, different nitrogen transformation mechanisms at different SGD sites, possibly associated with different geology.) which could provide a contribution to the literature. A major limitation to the study is the number of samples analyzed. Authors obtained and analyzed one sample from each site during one sampling period although this study would clearly benefit from expanded sampling to make the conclusions from convincing. Given that analysis error and potential biases from sampling, for example, it is not sure whether the general characteristics of microbial composition, distribution and its function of each site can be reasonably represented. Most of chemical data presented does not include statistical information such as error range, mean, p-values, etc. The PCA analysis result revealed that salinity and nitrate are key environmental factors controlling microbial community. Unfortunately, however, authors did not get salinity data for some key samples (TS-4, 5), and referenced the data from previous studies. Given that the significant effect of salinity on the analysis results, this issues needs to be tested more carefully. First, authors needs to show that the hydrological condition between two sampling period were similar, thus there is limited seasonal (temporal) effect in the salinity profile. Second, the authors has to present the salinity data from TS 1-3 from pervious field campaign and confirm that the shallow porewater (TS1-3) presented low salinity (freshwater) values during both field campaign period.

Manuscript needs moderate revision for its structure. I found it is difficult to follow the manuscript in parts, which are common when many things were summed up in one paper.

(1) Line 97:

The SGD and nutrient fluxes better to be represented as m/day (mmol/m2/day) or m3/m/day (mmol/m/day). Considering the different shoreline length and size of two sites, SGD fluxes shown in m3/day does not provide adequate information here. Also, please try to add error range of estimated SGD (nutrient flux).

(2) Line 116: MB1-6 -> MB1-7??

(3) Please consider switching the order of Figure 1 and 2. Also, please mark the location of vertical profiles of Fig 1 into the Fig. 2. The sampling depth betters to be clearly shown in Fig 1, so that authors can compare the monitoring depth and its geological formation at the same time.

(4) Line 168: As explained above, provide more detailed and clear information concerning the previous sampling period. Additionally, please check show that the hydrological condition between two sampling period were similar and the salinity data from TS 1-3 from pervious field campaign showed similar values with current field survey.

(5) Table 1. Provide additional statistical information (error bar, mean, p-values) if possible. Additionally, kindly check data for NO3-, NH4+, PO4 and DON for TS-SE-4 and TS-SE-5 (All data are same each other).

(6) Line 254: Based on the Fig 7, the difference microbial community between TS-SE3/4 and TS-SE-5 seems to be associated with different geological layer. It would be better if authors can clarify the geological difference in each sampling point in the beginning so that authors can better understand the link between microbial composition and geologic layer.

(7) Figure resolution is too low. Some tests in figures are too small (e.g. coordinate information in fig 2). Please enhance the figure resolution and increase the text sizes.

Reviewer #2: Revision of the manuscript “Microbial community composition across a coastal hydrological system affected by submarine groundwater discharge (SGD)” by Adyasari et al.

The paper of Adyasari et al. explores the spatial dynamics of prokaryotic communities along a coastal hydrological gradient in the Mobile Bay, USA, which is influenced by submarine groundwater discharge (SGD), as well as its links to environmental conditions. To do so, they collect water samples from 20 sites, including fresh groundwater, two vertical profiles of porewater in the intertidal zones, river water and bay water. The bacterial community composition is characterized by sequencing of the 16S rRNA gene, and the physicochemical parameters measured include salinity, temperature, pH, dissolved oxygen and concentration of inorganic nutrients. In addition, they estimate the functional diversity from the taxonomic data using the Tax4Fun2 method.

Their results show pronounced changes in the structure of the prokaryotic assemblages between the different habitats covered, and a high vertical heterogeneity within each piezometer. The environmental variables most strongly related to the observed variations in microbial communities are salinity and nitrate, and they find higher diversity in the intertidal pore waters than in the fresh groundwater and in surface river or seawater. The authors provide a detailed and complete description of the taxonomic groups detected. Based on the functional analysis, the authors suggest that subsurface microbial communities may play a large role in nitrogen cycling, thus affecting the nutrients delivered to the bay. The authors conclude that that their results support previous studies in the area showing that SGD supports productivity.

The research presented here is relevant because it provides information about the spatial variations in microbial communities in subterranean estuaries, largely undersampled for microbial studies in comparison to other surface aquatic habitats. The study is also of relevance since it compares the original groundwater, river water, bay water and pore waters located in a single area, which allows a comprehensive exploration of the microbial diversity in this groundwater-marine interface, which is also one of the least studied ecotones in terms of microbial ecology and biogeography. The study shows how complex and heterogeneous groundwater microbial assemblages are, even at such small spatial scales. Given that groundwater aquifers appear to hide a vast diversity of unknown taxa, any study providing insight on the subsurface microbial inhabitants is of interest. The manuscript is well written, the results are well presented and clearly explained, and the discussion is well contextualized.

In my opinion, however, the study needs to address some issues, and to acknowledge some of its limitations. For example, this type of intertidal systems may be extremely variable, not only depending on tidal fluctuations but also on hydrologic regimes. As this paper presents a single snap-shot of the system (one sampling day), the conclusions must be taken with caution and the existence of large temporal variations in the reported patterns at different scales must be acknowledged and discussed in the Discussion section. In my opinion, the manuscript should also better highlight the lack of studies targeting groundwater estuaries, and should put their results in the context of such previous knowledge; although to my knowledge very few studies have explored the microbiology in these type of systems, there are some that have not been included in the manuscript and thus should be cited (see below). Also, the authors find important proportions of Archaea, and to my knowledge not much is known about archaea in groundwaters. A better contextualization of these results in view of the existing literature would render the manuscript more interesting and novel, as there is truly a lack of studies focusing on these sites that appear to be important hotspots of biogeochemical activity.

Given the large spatial heterogeneity reported in the vertical pore water profiles, I think that it would be interesting to address the drivers of the studied communities including only the subsurface samples, as it is expectable that salinity will be one of the drivers explaining the majority of the variation when river and bay samples are included. However, based on the NMDS and the PCA, it does not seem that salinity is the main driver of the variations across/within the subsurface samples, is it? I suggest that the analysis is performed in parallel including all samples together but also considering only the subsurface ones.

Finally, as the sampling covers a nice groundwater-river-brackish gradient, I think it would be interesting to explore a little bit more whether the authors can find indicator taxa associated to a given habitat, or to which extent the different ground- and porewater samples are comprised of typical marine or typical freshwater taxa (see comments below).

Although I see that this manuscript provides interesting and detailed information of the studied assemblages, I recommend that these and other issues (see below) are addressed for this study to be published in PLOS ONE.

Major comments:

1. Overall, I see the relevance of the research and the interest of the sampling design, but in my opinion the manuscript would largely benefit from modifying partially the Introduction and the Discussion so that the relevance, novelty or need of conducting the present research is better contextualized and justified. The study of groundwater microbial ecology is gaining increasing attention, and there have been many recent papers on groundwater communities, including some targeting subterranean estuaries (e.g. Héry et al. 2014 FEMS, Chen et al. 2019 STOTEN, Sang et al 2018 Sci Reports). A comparison to such previous studies would highlight the similarities or differences with such previous investigations, thus underlying the specific contributions of this investigation.

2. As mentioned before, the authors have interesting information that is however not shown in the figures. For example, could they make a figure analogous to Figure 3 and Figure 6 but considering only the subsurface communities? This would help better identify the environmental gradients that drive the pronounced spatial heterogeneity found within and across these communities. Including the river and bay water is interesting to place subsurface communities within the whole hydrologic gradient, but it is well known that salinity drives bacterial communities in the surface fresh-marine transitions... at the subsurface, this may be less clear and assemblages be strongly influenced by other gradients such as dissolved oxygen, which will be more clearly visualized when only those samples are considered.

3. Similarly, the fact that the surface seawater and freshwater are also included allows for a deeper exploration of e.g. typical groundwater indicators (indval analysis), or to explore the percentage of 'river' or 'marine' taxa that comprise the studied subsurface communities. There is also a recent study (Rocca et al 2020 Ecology) that is relevant in this regard.

Specific comments:

-One of the main conclusions stated in the abstract is not easily derived from the results reported in the abstract: 'Overall, our study supported previous observations in the area highlighting the role of SGD in Mobile Bay's productivity'. This does not seem to be the main conclusion of the paper as nothing has been shown regarding the productivity in the Bay. I suggest that the end of the abstract is rewritten to deliver a more specific message that can be directly derived from the presented results.

-It is not clear why so many different bay water samples were taken. What was the purpose of this? Are they supposed to cover sites of different SGD influence? Based on temperature, for example, it seems that they are quite different in terms of salinity... something should be said about this in the methods section (to justify the choice of bay sites).

-Line 110: Please indicate how was GW1 collected. Is this groundwater sample supposed to be indicative of the groundwater in the area? Please indicate so.

-Line 127: How where the duplicate samples treated for the different analyses? Where they averaged? Please explain it. The duplicates are not shown in the figures, and I think it would be interesting to visually see whether the individual replicates cluster together in the ordination graphs

-Line 221-223: I find very interesting the large variations in taxonomic richness and diversity found within the TS-SE vertical profile. Something should be said about this. Where these changes in taxonomic diversity explained by any of the measured environmental parameters?

-Line 272: The figures should be explained in the order of appearance, so the authors should either redo the figures or reorganize the text. In my opinion, the general composition at the class level should be explained together with the explanation of Figure 5, focusing on the main differences between river, bay water and groundwater assemblages. Then, the current subsection (line 272) could focus exclusively on the vertical dimension, adding, as suggested above, the composition in terms of order, or family, besides genera. Both things could be later combined in the Discussion, but the results section needs to follow the figure order. Otherwise it is confusing for the reader.

-Table 1. Please check the nutrient values for TS-SE4 and 5, and TS-NE1 and 2, as they are exactly the same.

-Figure 3. As suggested above, I think it would be interesting to look at these patterns considering only the subsurface samples

-Lines 237-246: This figure includes the taxonomic composition but nothing is said about it, only the Bray-Curtis dissimilarity is mentioned. Please describe the compositional patterns shown in the figure. The description of the figures should follow the sequence of appearance.

-Figure 7: if I understand it correctly, the grey area lumps together all the taxa that were not classified at the genus level. However, as this area is so big, wouldn't it be possible to assign it to the smaller taxonomic level possible in those cases? e.g. family.... Alternatively, I think it would be useful to draw the same profile in parallel to this one, grouping taxa at the e.g. order level, in order to differentiate it from Figure 5 but to show more visually the vertical variations in composition (which are not as clearly shown in Figure 5 because sites are clustered based on similarity and not along the vertical profile).

In other words, I suggest that you add a similar graph to the left of figure 7 that shows the whole taxonomic composition at a higher taxonomic rank (something in between the class level shown in Figure 5 and the genus level shown in Figure 7). This will help visualizing more clearly the vertical changes in composition, and then you can go to the identified genera in more detail.

-Figure 7: Here and throughout the text, replace on genus/class level by 'at the genus/class level'

-Figure 8: The proportion is calculated relative to what? to the total identified KOs? Please specify in the legend.

-Line 272: Microbial community composition at the horizontal and vertical scales?

-Line 272: The figures should be explained in the order of appearance, so the authors should either redo the figures or reorganize the text. In my opinion, the general composition at the class level should be explained together with the explanation of Figure 5, focusing on the main differences between river, bay water and groundwater assemblages. Then, the current subsection (line 272) could focus exclusively on the vertical dimension, adding, as suggested above, the composition in terms of order, or family, besides genera. Both things could be later combined in the Discussion, but the results section needs to follow the figure order. Otherwise it is confusing for the reader.

DISCUSSION: As explained above, in my opinion the manuscript would also benefit from expanding the discussion in order to compare the reported data with what has been done in previous studies targeting comparable systems. Although (at least to my knowledge) there are very few investigations exploring groundwater aquifers in coastal sites, these should be acknowledged and the results should be compared with them, in order to highlight the differences or similarities in the reported patterns (e.g. Héry et al. 2014 FEMS, Chen et al. 2019 STOTEN, Sang et al 2018 Sci Reports).

- Line 354: Changes in microbial community structure along the freshwater-marine continuum?

-Lines 364-372: This statement could be nicely shown if exploring the contribution to each subsurface community of typical groundwater, river water or marine water taxa. I expect that this would nicely show what are the mixing zones, and which are those sites where totally different communities arise, that do not result from the passive mixing of marine and freshwater taxa. There is accumulating evidence that microbial communities are largely structured by hydrologic connectivity (e.g. Ruiz-González et al 2015 Ecol Lett shows that freshwater bacterial communities are dominated by rare taxa transported from soils, some of which appear to grow in water). Rocca et al (2020 Ecology) is also relevant here ('Rare microbial taxa emerge when communities collide: freshwater and marine microbiome responses to experimental mixing'). There are other papers showing that the resuscitation of rare taxa can impact ecosystem processes such as CO2 and methane production (Aanderud et al. 2015 Front. Microbiol, Stegen et al. 2015 Nature Communications), so that the rare taxa that appear in the studied pore water sites may play large roles in coastal biogeochemistry. The manuscript would benefit from an expansion of the discussion on this topic, on the role of hydrologic transport of microbial diversity and the selective growth of some of the transported taxa when encountering new environmental conditions.

-Lines 432-440: To my knowledge, not much is known about Archaea in subsurface ecosystems, and in particular in subterranean estuaries. The authors should highlight what is known or not about the topic, in order to better highlight the specific contributions of the presented research.

-Line 444-445. Please rewrite this sentence to specify more clearly that NO3 is transformed to NH4 during transit through this site, which is then delivered to the Bay

-Line 446, replace 'and it is likely generated' with ',which are likely generated'

-Line 450: was related

-Line 458: depending on

REFERENCES CITED HERE

Aanderud, Z.T., Jones, S.E., Fierer, N. & Lennon, J.F. (2015) Resuscitation of the rare biosphere contributes to pulses of ecosystem activity. Frontiers in Microbiology, 6, 24.

Héry, M., Volant, A., Garing, C., Luguot, L., Poulichet, F.E. & Gouze, P. (2014) Diversity and geochemical structuring of bacterial communities along a salinity gradient in a carbonate aquifer subject to seawater intrusion. Fems Microbiology Ecology, 90, 922–934.

Rocca, J.D., Simonin, M., Bernhardt, E.S., Washburne, A.D. & Wright, J.J. (2020) Rare microbial taxa emerge when communities collide: freshwater and marine microbiome responses to experimental mixing. Ecology, 101, e02956.

Ruiz-González, C., Niño-García, J.P. & del Giorgio, P.A. (2015) Terrestrial origin of bacterial communities in complex boreal freshwater networks. Ecology Letters, 18, 1198-1206.

Sang, S., Zhang, X., Dai, H., Hu, B.X., Ou, H. & Sun, L. (2018) Diversity and predictive metabolic pathways of the prokaryotic microbial community along a groundwater salinity gradient of the Pearl River Delta, China Scientific Reports, 8, 17317.

Stegen, J.C., Fredrickson, J.K., Wilkins, M.J., Konopka, A.E., Nelson, W.C., Arntzen, E.V., Chrisler, W.B., Chu, R.K., Danczak, R.E., Fansler, S.J., Kennedy, D.W., Resch, C.T. & Tfaily, M. (2016) Groundwater–surface water mixing shifts ecological assembly processes and stimulates organic carbon turnover. Nature Communications, 7, 11237.

6. PLOS authors have the option to publish the peer review history of their article (what does this mean?). If published, this will include your full peer review and any attached files.

Reviewer #1: No

Reviewer #2: No

---

## [Author Response · Author response to Decision Letter 0]

9 Jun 2020

We thank Reviewer 1 and 2 for their constructive feedbacks. We would also acknowledge that there have been three differences between this revised manuscript with the original submission: 

a. There has been a mistake on DON concentration of pore water in the original submission. We have corrected this mistake in this revised submission. 

b. There has been a mistake on coordinate of TS-NE-3 in the original submission, as this sampling point was a standalone piezometer and not a part of TS-NE vertical profile. We corrected this mistake in this revised submission. 

c. Considering Reviewer 1 concern related to the unavailability of salinity, DO, and temperature data for some pore water samples, and suggestion from Reviewer 2 about separating subsurface microbial samples, we modified our hypothesis testing method from redundancy analysis (RDA) to permutational multivariate analysis of variance (PERMANOVA) applied to surface and subsurface samples separately, instead of applying the test to all samples altogether. Following the suggestion of Reviewer 2, we explored the relationships between environmental parameters separately for subsurface samples, and noticed that the correlation between environmental parameter in surface and subsurface samples was fundamentally different. Therefore, we proceeded with separate analyses for surface and subsurface samples. PERMANOVA was chosen as a more robust alternative to RDA. 

Reviewer 1

Reviewer #1: SGD has been recognized as a major pathway of inland-derived nutrients and chemical flux into ocean. Nitrogen transformation mediated by chemical and microbial reaction along its pathway from inland aquifer to subterranean estuary could serve as a major mechanism controlling the level and form of nutrient discharge into the ocean. This paper aims to investigate microbial composition and distribution along the coastal subsurface/surface water bodies influenced by nutrient-rich SGD. The manuscript presents interesting results (sharp shifts in microbial composition depending on location and environmental factors, different nitrogen transformation mechanisms at different SGD sites, possibly associated with different geology.) which could provide a contribution to the literature. A major limitation to the study is the number of samples analyzed. Authors obtained and analyzed one sample from each site during one sampling period although this study would clearly benefit from expanded sampling to make the conclusions from convincing. Given that analysis error and potential biases from sampling, for example, it is not sure whether the general characteristics of microbial composition, distribution and its function of each site can be reasonably represented. 

We thank the reviewer for their constructive comments and we try to answer their concerns in more detailed manners in the points below. To address Reviewer 1 concern about the one-time sampling period, we acknowledge this limitation and have stated in the manuscript that further studies are needed to characterize the microbial community composition and its environmental drivers in higher certainty, for instance by measuring more environmental parameters (line 444-445) or taking samples over longer period of time, i.e. covering different tidal cycle and seasons (line 526-528). This study provides first insight of subsurface microbial activities in Mobile Bay area. Considering that the result of this study points to active nitrogen and methane cycles in the STE, it may serve as a primer to more detailed microbial studies on nitrogen and methane in the future, particularly in view of their contributions to water quality and climate variability in Mobile Bay (i.e. ammonium flux as the main cause of Jubilees and methane as a major contributor of climate change) or northern Gulf of Mexico area with similar lithological settings. 

Most of chemical data presented does not include statistical information such as error range, mean, p-values, etc. The PCA analysis result revealed that salinity and nitrate are key environmental factors controlling microbial community. Unfortunately, however, authors did not get salinity data for some key samples (TS-4, 5), and referenced the data from previous studies. Given that the significant effect of salinity on the analysis results, this issues needs to be tested more carefully. First, authors needs to show that the hydrological condition between two sampling period were similar, thus there is limited seasonal (temporal) effect in the salinity profile. Second, the authors has to present the salinity data from TS 1-3 from pervious field campaign and confirm that the shallow porewater (TS1-3) presented low salinity (freshwater) values during both field campaign period. 

To address the concern related data availability for salinity, DO, and temperature, we present detailed hydrological data from the previous sampling expeditions in one of the answer point below. However, as the previous expeditions were not conducted as detailed as our sampling (i.e. taking samples in 0.5-1.5 m intervals along the vertical profiles), some assumptions were still made to fulfill the requirement for PCA and envfit analyses. Our basis of filling the empty columns is based on two-year observation from Montiel, Lamore (1), Montiel, Lamore (2). For example, from other TS-NE measurement where more than one sample was collected (July 2016), parameters such as temperature, salinity, and DO were almost uniform throughout the depth; thus, we also used this assumption to fill the empty July 2017 data. Thus, we used this assumption to fill the empty columns in this study. Nevertheless, we do not use these parameters (salinity, temperature, DO) anymore as environmental predictors of community shifts among the subsurface samples in the PERMANOVA, to avoid any bias associated with estimating the missing values.

Manuscript needs moderate revision for its structure. I found it is difficult to follow the manuscript in parts, which are common when many things were summed up in one paper.

We revised the structure of the paper to be more streamlined and easier to follow.

(1) Line 97: The SGD and nutrient fluxes better to be represented as m/day (mmol/m2/day) or m3/m/day (mmol/m/day). Considering the different shoreline length and size of two sites, SGD fluxes shown in m3/day does not provide adequate information here. Also, please try to add error range of estimated SGD (nutrient flux).

We changed SGD and nutrient fluxes unit into m d-1 and mmol m-2 d-1, respectively, and displayed these results as range instead of average number. The revised version of this information can be found in line 105-116. 

(2) Line 116: MB1-6 -> MB1-7??

We corrected this sentence. 

(3) Please consider switching the order of Figure 1 and 2. Also, please mark the location of vertical profiles of Fig 1 into the Fig. 2. The sampling depth betters to be clearly shown in Fig 1, so that authors can compare the monitoring depth and its geological formation at the same time.

We removed Figure 2 and incorporated vertical depth and lithological profile of TS-SE and TS-NE into Figure 1. Figure 1 now depicts overview of study site, sampling points, and vertical lithological profile from two SGD sites. 

(4) Line 168: As explained above, provide more detailed and clear information concerning the previous sampling period. Additionally, please check show that the hydrological condition between two sampling period were similar and the salinity data from TS 1-3 from pervious field campaign showed similar values with current field survey.

We list the result of previous and the latest field campaigns in the table below, which can also be seen in Montiel, Lamore (1), Montiel, Lamore (2). Overall, our expedition in July 2017 covered more detailed sample collection and measurement than the previous campaigns, which did not take vertical profile into account. For example, from other TS-NE measurement where more than one sample was collected (July 2016), parameters such as temperature, salinity, and DO were almost uniform throughout the depth; thus, we also used this assumption to fill the empty July 2017 data. There is only one TS-SE measurement conducted in more than one layer aside of July 2017, i.e. December 2016, where temperature and salinity were similar, but DO concentration decreased with greater depth. We also applied this similar pattern into the empty data on July 2017. 

Time/season Sample Temperature (oC) Salinity DO (mg L-1)

June 2016 (dry season) TS-SE-B (only one measurement) 25.6 5.3 0.8

December 2016 (wet season) TS-SE-B1 (upper) 20.8 14.4 1.6

 TS-SE-B3 (mid) 20.7 14.5 1

March 2017 (dry season) TS-SE-B (only one measurement) 19.8 7.6 0.6

July 2017 (dry season, this manuscript) TS-SE-B1 29.9 0 0.6

 TS-SE-B2 29.4 1.8 - (0.6)

 TS-SE-B3 29.4 1.8 - (0.0)

 TS-SE-B4 - (29.4) - (1.8) - (0.0)

 TS-SE-B5 - (29.4) - (1.8) - (0.0)

April 2016 (dry season) TS-NE-A (only one measurement) 20.6 0.3 2.9

June 2016 (dry season) TS-NE-A1 27.9 0.1 1.3

 TS-NE-A2 28.1 0.1 1.4

March 2017 (dry season) TS-NE-A (only one measurement) 20.6 0.1 1

July 2017 (dry, this manuscript) TS-NE-A1 28.4 0.1 0.3

 TS-NE-A2 - (28.4) - (0.1) - (0.3)

Note: (-) represents empty data, while the number in brackets represents numbers that we use to fill the empty data

It has to be noted that we only used the estimated values in descriptive data visualizations, i.e. PCA (Figure 1) and envfit (Figure 5). For hypothesis testing, i.e. finding the strongest predictor for community shift in surface and subsurface samples, we did not use parameters with missing (and then estimated) values. Instead, we rely on the correlation between observed measurements of environmental parameters to interpret the results of the hypothesis test and infer the importance of the parameters with missing values. As mentioned before, PERMANOVA was used as a method more robust to violations of the assumptions of the parametric RDA.

(5) Table 1. Provide additional statistical information (error bar, mean, p-values) if possible. Additionally, kindly check data for NO3-, NH4+, PO4 and DON for TS-SE-4 and TS-SE-5 (All data are same each other).

We corrected the DON data in Table 1. The values reported in the table are for one sample per row. Therefore no means and error bars are provided. The measurements of the environmental parameters are mainly descriptive and were not designed for hypothesis testing. 

(6) Line 254: Based on the Fig 7, the difference microbial community between TS-SE3/4 and TS-SE-5 seems to be associated with different geological layer. It would be better if authors can clarify the geological difference in each sampling point in the beginning so that authors can better understand the link between microbial composition and geologic layer.

We explained the lithological condition in the original manuscript (line 99-104) and additionally, we put more information about organic matter content in this revised version. The revised sentence related to lithological condition of both SGD sites can be found in line 108-112 and line 114-116, while its visualization can be found in Figure 1C.

(7) Figure resolution is too low. Some tests in figures are too small (e.g. coordinate information in fig 2). Please enhance the figure resolution and increase the text sizes.

We enhanced the figure resolution and increased the text sizes. 

Reviewer 2

Reviewer #2: Revision of the manuscript “Microbial community composition across a coastal hydrological system affected by submarine groundwater discharge (SGD)” by Adyasari et al.

The paper of Adyasari et al. explores the spatial dynamics of prokaryotic communities along a coastal hydrological gradient in the Mobile Bay, USA, which is influenced by submarine groundwater discharge (SGD), as well as its links to environmental conditions. To do so, they collect water samples from 20 sites, including fresh groundwater, two vertical profiles of porewater in the intertidal zones, river water and bay water. The bacterial community composition is characterized by sequencing of the 16S rRNA gene, and the physicochemical parameters measured include salinity, temperature, pH, dissolved oxygen and concentration of inorganic nutrients. In addition, they estimate the functional diversity from the taxonomic data using the Tax4Fun2 method.

Their results show pronounced changes in the structure of the prokaryotic assemblages between the different habitats covered, and a high vertical heterogeneity within each piezometer. The environmental variables most strongly related to the observed variations in microbial communities are salinity and nitrate, and they find higher diversity in the intertidal pore waters than in the fresh groundwater and in surface river or seawater. The authors provide a detailed and complete description of the taxonomic groups detected. Based on the functional analysis, the authors suggest that subsurface microbial communities may play a large role in nitrogen cycling, thus affecting the nutrients delivered to the bay. The authors conclude that that their results support previous studies in the area showing that SGD supports productivity.

The research presented here is relevant because it provides information about the spatial variations in microbial communities in subterranean estuaries, largely undersampled for microbial studies in comparison to other surface aquatic habitats. The study is also of relevance since it compares the original groundwater, river water, bay water and pore waters located in a single area, which allows a comprehensive exploration of the microbial diversity in this groundwater-marine interface, which is also one of the least studied ecotones in terms of microbial ecology and biogeography. The study shows how complex and heterogeneous groundwater microbial assemblages are, even at such small spatial scales. Given that groundwater aquifers appear to hide a vast diversity of unknown taxa, any study providing insight on the subsurface microbial inhabitants is of interest. The manuscript is well written, the results are well presented and clearly explained, and the discussion is well contextualized.

In my opinion, however, the study needs to address some issues, and to acknowledge some of its limitations. For example, this type of intertidal systems may be extremely variable, not only depending on tidal fluctuations but also on hydrologic regimes. As this paper presents a single snap-shot of the system (one sampling day), the conclusions must be taken with caution and the existence of large temporal variations in the reported patterns at different scales must be acknowledged and discussed in the Discussion section. 

We thank Reviewer 2 for their constructive comments and we try to answer their concerns in more detailed manners in the points below. We acknowledge the limitation of single sampling and state that further studies are needed to characterize the microbial community composition and its environmental driver in higher certainty by measuring more environmental parameters (line 444-445) or taking samples over a longer period of time, i.e. covering different tidal cycle and seasons (line 526-528). This study provides first insight into subsurface microbial communities, and considering that the result of this study points to active nitrogen and methane cycles in the STE, it may serve as a primer for more detailed microbial studies on nitrogen and methane in the future, particularly in view of their contributions to water quality and climate variability in Mobile Bay (i.e. ammonium flux as the main cause of Jubilees and methane as a major contributor of climate change) or northern Gulf of Mexico area with similar lithological settings.

In my opinion, the manuscript should also better highlight the lack of studies targeting groundwater estuaries, and should put their results in the context of such previous knowledge; although to my knowledge very few studies have explored the microbiology in these type of systems, there are some that have not been included in the manuscript and thus should be cited (see below). 

We thank the reviewer for the references given to improve our manuscript. We explained the limited subsurface microbial studies relative to microbial studies conducted in surface water in the original manuscript (line 75-80). We did not discuss microbial communities of groundwater-to-estuaries in this manuscript because the majority of groundwater flows in the study site lead to coastal water instead of estuaries. Nevertheless, we added discussion about the comparison of result between our study and other previous studies in similar settings (line 415-445, line 512-528, line 584-587). 

Also, the authors find important proportions of Archaea, and to my knowledge not much is known about archaea in groundwater. A better contextualization of these results in view of the existing literature would render the manuscript more interesting and novel, as there is truly a lack of studies focusing on these sites that appear to be important hotspots of biogeochemical activity.

We added archaea-related information in the abstract (line 29-31), introduction (line 62-65), result (line 386-395), and discussions (line 545-558, line 602-612). Generally, we found that the dominant archaeal taxa found in TS-SE were commonly found in other reducing environment and associated with methane cycle, which was also supported the result of metabolic prediction of methane in our study. We also found archaea in TS-NE, despite low relative proportion in comparison to TS-SE. Nevertheless, the archaeal taxa identified in TS-NE were potentially contributed to ammonia/methane oxidation in this site based on the metabolic prediction. 

Given the large spatial heterogeneity reported in the vertical pore water profiles, I think that it would be interesting to address the drivers of the studied communities including only the subsurface samples, as it is expectable that salinity will be one of the drivers explaining the majority of the variation when river and bay samples are included. However, based on the NMDS and the PCA, it does not seem that salinity is the main driver of the variations across/within the subsurface samples, is it? I suggest that the analysis is performed in parallel including all samples together but also considering only the subsurface ones.

We thoroughly revised our approach for the statistical analyses, i.e. separating surface and subsurface samples, as well as better accounting for the missing values in the subsurface samples. In this revision, PCA, NMDS and PERMANOVA were conducted each for surface and subsurface samples. Of the observed parameters in this study, salinity was found as the strongest predictor of surface water community shift, while DON and PO43- were identified as the most determinant variable explaining pattern in subsurface microbial community composition (line 310-321, Table 2). While we did not use salinity, DO, and temperature as variables in PERMANOVA analysis, the selection of DON and PO43- as significant predictors implied a potentially similarly important role of DO and salinity, as any effect attributed to DON and PO43- could also be explained by DO and salinity due to their high correlation (DON and DO: Pearson r = -0.74, PO43- and salinity: Pearson r = 0.98). 

Finally, as the sampling covers a nice groundwater-river-brackish gradient, I think it would be interesting to explore a little bit more whether the authors can find indicator taxa associated to a given habitat, or to which extent the different ground- and porewater samples are comprised of typical marine or typical freshwater taxa (see comments below).

We did not perform indicator taxa analysis (indval) as our sampling design was not suitable for such an analysis. Nevertheless, we conducted analysis on taxa with the highest proportion and best fitted to environmental parameter with function ordiselect from ‘goeveg’ R package in the original submission (line 281-331 in the original submission discusses the majority of taxa resulted from this analysis). Cyanobium PCC-6307 and unclassified Thermodesulfovibrionia were identified as dominant taxa strongly associated with trends in environmental parameters in surface and subsurface samples, respectively. The other taxa resulted from ordiselect analysis are displayed in Figure 5 relative to sampling location and environmental parameters.

Although I see that this manuscript provides interesting and detailed information of the studied assemblages, I recommend that these and other issues (see below) are addressed for this study to be published in PLOS ONE.

We thank the reviewer for their constructive comments and appreciate the references given by the reviewer. We try to answer in details their concerns in the points below.

Major comments:

1. Overall, I see the relevance of the research and the interest of the sampling design, but in my opinion the manuscript would largely benefit from modifying partially the Introduction and the Discussion so that the relevance, novelty or need of conducting the present research is better contextualized and justified. The study of groundwater microbial ecology is gaining increasing attention, and there have been many recent papers on groundwater communities, including some targeting subterranean estuaries (e.g. Héry et al. 2014 FEMS, Chen et al. 2019 STOTEN, Sang et al 2018 Sci Reports). A comparison to such previous studies would highlight the similarities or differences with such previous investigations, thus underlying the specific contributions of this investigation.

We explained the research gaps related to the limited subsurface microbial studies in comparison to surface water studies in the original manuscript (line 75-77). In this revised manuscript, we added further information in Introduction and Discussion based on some references suggested by Reviewer 2. We discussed and compared our result related to significant predictor of community shift with other similar studies (line 415-445). We discussed the potential of community coalescence in fresh-,marine water as discussed by Rocca, Simonin (3) (line 512-528). We also added a discussion about low proportion taxa potentially driving the dominant elemental cycle in STE, as also discussed by Héry, Volant (4) (line 602-612). 

2. As mentioned before, the authors have interesting information that is however not shown in the figures. For example, could they make a figure analogous to Figure 3 and Figure 6 but considering only the subsurface communities? This would help better identify the environmental gradients that drive the pronounced spatial heterogeneity found within and across these communities. Including the river and bay water is interesting to place subsurface communities within the whole hydrologic gradient, but it is well known that salinity drives bacterial communities in the surface fresh-marine transitions... at the subsurface, this may be less clear and assemblages be strongly influenced by other gradients such as dissolved oxygen, which will be more clearly visualized when only those samples are considered.

We revised some of the statistical analyses in this manuscript, i.e. separating surface and subsurface samples to accommodate the missing values in the subsurface samples. In this revision, PCA, NMDS and PERMANOVA were conducted each for surface and subsurface samples. PERMANOVA result indicates that out of observed parameters in this study, salinity was found as the strongest predictor of surface water community shift, while DON and PO43- were identified as the most determinant variable explaining pattern in subsurface microbial community composition (line 310-321, Table 2). Nevertheless, DON and PO43- only contributed 25% to total explainable variance, indicating there are other variables that are stronger predictors than our measured parameters. This could be the concentration of various electron acceptors (e.g. SO42- or CH4, as we identified many SRBs and methanogens in TS-SE), ion concentration, or tidal fluctuation. This explanation can be found in line 415-445. 

3. Similarly, the fact that the surface seawater and freshwater are also included allows for a deeper exploration of e.g. typical groundwater indicators (indval analysis), or to explore the percentage of 'river' or 'marine' taxa that comprise the studied subsurface communities. There is also a recent study (Rocca et al 2020 Ecology) that is relevant in this regard.

We did not perform indval analysis as our sampling design was not suitable to reliably predict indicator OTUs. Instead we identified dominant taxa which were best fitted to the environmental parameters using ordiselect function, which shows that Cyanobium PCC-6307 and unclassified Thermodesulfovibrionia were identified as the “representative” OTUs for surface and subsurface samples, respectively. We describe the patterns in the contribution of these taxa to the microbial community composition in the different sample groups in line 339-343 and line 362-366. 

In general, we did not observe any significant occurrence of community coalescence between freshwater and marine bacteria in samples, unlike the study done by Rocca, Simonin (3). For example, freshwater taxa were found in both groundwater and river, but not in Mobile Bay water column which was dominated by marine bacteria, at least not in distinguishable proportion. The only exception was the occurrence of marine taxa in samples from river and the top layer of pore water, which were most likely brought to estuaries and upper STE by the previous high tide. The prevalence of marine taxa in freshwater, but not the opposite, is attributable to the fact that marine taxa have a higher resistance to salinity changes in comparison to freshwater taxa (5). This discussion can be found in line 512-528. 

Specific comments:

- One of the main conclusions stated in the abstract is not easily derived from the results reported in the abstract: 'Overall, our study supported previous observations in the area highlighting the role of SGD in Mobile Bay's productivity'. This does not seem to be the main conclusion of the paper as nothing has been shown regarding the productivity in the Bay. I suggest that the end of the abstract is rewritten to deliver a more specific message that can be directly derived from the presented results.

We corrected the abstract and conclusion related to this point.

- It is not clear why so many different bay water samples were taken. What was the purpose of this? Are they supposed to cover sites of different SGD influence? Based on temperature, for example, it seems that they are quite different in terms of salinity... something should be said about this in the methods section (to justify the choice of bay sites).

Samples from Mobile Bay water column were collected from different part of the bay to cover different salinity and potential influence of river to microbial community composition at the bay: samples MB1-4 were collected in the southern part of Mobile Bay where the influence of river was almost negligible, while MB6-7 were taken close to estuaries in the northern part of the bay. This added explanation can be found in line 140-143. 

- Line 110: Please indicate how GW1 was collected. Is this groundwater sample supposed to be indicative of the groundwater in the area? Please indicate so.

The water sample for GW-1 was collected after pumping the well with a submersible pump at a constant rate until conductivity, temperature, and DO values were stable. We chose this sampling point is representative of the terrestrial groundwater based on result from Montiel, Lamore (2). This added explanation on sample collection from can be found in line 143-145.

- Line 127: How where the duplicate samples treated for the different analyses? Where they averaged? Please explain it. The duplicates are not shown in the figures, and I think it would be interesting to visually see whether the individual replicates cluster together in the ordination graphs.

This is our mistake as we took duplicate samples; however, only one set of sample was sequenced for 16S rRNA. We deleted the “duplicate” in the manuscript. 

- Line 221-223: I find very interesting the large variations in taxonomic richness and diversity found within the TS-SE vertical profile. Something should be said about this. Where these changes in taxonomic diversity explained by any of the measured environmental parameters?

The result of alpha diversity of TS-SE vertical profile is shown in line 263-268 (Result) and line 448-455 (Discussion). Generally, the highest Shannon and inverse Simpson indices across all samples was found in TS-SE-B2 and TS-SE-B3, which could be attributable to diverse mixture of fresh water and marine microbes, as well as aerobic, facultative anaerobic, anaerobic bacteria and archaea in the reduced layer of TS-SE, thus increasing microbial alpha diversity in this site. A Spearman correlation analysis between Shannon and inverse Simpson indices with observed environmental parameters resulted in DON and PO43- as variables with the highest correlation with the diversity indices (line 269-272 (result) and S1 Table). This finding was similar with the result from PERMANOVA analysis which indicated DON and PO43- as the main predictors for subsurface microbial community shift. 

- Line 272: The figures should be explained in the order of appearance, so the authors should either redo the figures or reorganize the text. In my opinion, the general composition at the class level should be explained together with the explanation of Figure 5, focusing on the main differences between river, bay water and groundwater assemblages. Then, the current subsection (line 272) could focus exclusively on the vertical dimension, adding, as suggested above, the composition in terms of order, or family, besides genera. Both things could be later combined in the Discussion, but the results section needs to follow the figure order. Otherwise it is confusing for the reader.

We re-organized the text and figures in the manuscript to be more streamlined and easier to follow. In this revised version, we chose to explain the result of the microbial diversity and statistical result first (line 243-321, Figure 2-5), then proceed to the description of microbial community composition in class and genus level (line 329-395), which we put together their bar plots in Figure 6. 

- Table 1. Please check the nutrient values for TS-SE4 and 5, and TS-NE1 and 2, as they are exactly the same.

We corrected the mistype in Table 1.

- Figure 3. As suggested above, I think it would be interesting to look at these patterns considering only the subsurface samples

We modified PCA plot to show both surface and subsurface environmental parameters (now Figure 2). 

- Lines 237-246: This figure includes the taxonomic composition but nothing is said about it, only the Bray-Curtis dissimilarity is mentioned. Please describe the compositional patterns shown in the figure. The description of the figures should follow the sequence of appearance.

We explained the Bray-Curtis dissimilarity in the original manuscript (line 237-246), now in line 279-289. 

- Figure 7: if I understand it correctly, the grey area lumps together all the taxa that were not classified at the genus level. However, as this area is so big, wouldn't it be possible to assign it to the smaller taxonomic level possible in those cases? e.g. family.... Alternatively, I think it would be useful to draw the same profile in parallel to this one, grouping taxa at the e.g. order level, in order to differentiate it from Figure 5 but to show more visually the vertical variations in composition (which are not as clearly shown in Figure 5 because sites are clustered based on similarity and not along the vertical profile). In other words, I suggest that you add a similar graph to the left of figure 7 that shows the whole taxonomic composition at a higher taxonomic rank (something in between the class level shown in Figure 5 and the genus level shown in Figure 7). This will help visualizing more clearly the vertical changes in composition, and then you can go to the identified genera in more detail.

To avoid redundancy of information, we decided to present the data only at class and genus level to provide a more general and a detailed overview of the microbial community composition, respectively. To avoid an unnecessary complexity of the figures, we only show the most dominant taxa, which are also relevant for the discussion. After careful consideration, we decided that the added value of showing more taxonomic levels would be minimal and would rather result in a loss of focus and conciseness of the manuscript. Additionally, the full OTU table and associated taxonomic classification are available as part of the public archived data on Pangaea (https://doi.pangaea.de/10.1594/PANGAEA.912763). We hope you understand our rationale for not changing the taxonomic resolution of the figures.

- Figure 7: Here and throughout the text, replace on genus/class level by 'at the genus/class level'

We replaced “on genus/class level” by “at the genus/class level” throughout the text and caption.

- Figure 8: The proportion is calculated relative to what? to the total identified KOs? Please specify in the legend.

We specified “Proportion of total identified KOs” in the legend.

- Line 272: Microbial community composition at the horizontal and vertical scales?

We revised this chapter sub-title into “Microbial community composition at the horizontal and vertical scales”.

- Line 272: The figures should be explained in the order of appearance, so the authors should either redo the figures or reorganize the text. In my opinion, the general composition at the class level should be explained together with the explanation of Figure 5, focusing on the main differences between river, bay water and groundwater assemblages. Then, the current subsection (line 272) could focus exclusively on the vertical dimension, adding, as suggested above, the composition in terms of order, or family, besides genera. Both things could be later combined in the Discussion, but the results section needs to follow the figure order. Otherwise it is confusing for the reader.

We re-organized the text and figures in the manuscript to be more streamlined and easier to follow. However, as explained previously we refrain from including a description at more taxonomic levels to avoid redundancy.

- DISCUSSION: As explained above, in my opinion the manuscript would also benefit from expanding the discussion in order to compare the reported data with what has been done in previous studies targeting comparable systems. Although (at least to my knowledge) there are very few investigations exploring groundwater aquifers in coastal sites, these should be acknowledged and the results should be compared with them, in order to highlight the differences or similarities in the reported patterns (e.g. Héry et al. 2014 FEMS, Chen et al. 2019 STOTEN, Sang et al 2018 Sci Reports).

We added more information in the Discussion, particularly the comparison of our result with other similar studies (line 415-445, line 512-528, line 584-587).

Line 354: Changes in microbial community structure along the freshwater-marine continuum?

We revised the chapter sub-title into “Changes in microbial community composition along the freshwater-marine water continuum”.

- Lines 364-372: This statement could be nicely shown if exploring the contribution to each subsurface community of typical groundwater, river water or marine water taxa. I expect that this would nicely show what are the mixing zones, and which are those sites where totally different communities arise, that do not result from the passive mixing of marine and freshwater taxa. There is accumulating evidence that microbial communities are largely structured by hydrologic connectivity (e.g. Ruiz-González et al 2015 Ecol Lett shows that freshwater bacterial communities are dominated by rare taxa transported from soils, some of which appear to grow in water). Rocca et al (2020 Ecology) is also relevant here ('Rare microbial taxa emerge when communities collide: freshwater and marine microbiome responses to experimental mixing'). There are other papers showing that the resuscitation of rare taxa can impact ecosystem processes such as CO2 and methane production (Aanderud et al. 2015 Front. Microbiol, Stegen et al. 2015 Nature Communications), so that the rare taxa that appear in the studied pore water sites may play large roles in coastal biogeochemistry. The manuscript would benefit from an expansion of the discussion on this topic, on the role of hydrologic transport of microbial diversity and the selective growth of some of the transported taxa when encountering new environmental conditions.

We did not sequence our samples deep enough to reliably assess rare taxa as the purpose of this study was assessing the dominant taxa; thus, our current data is insufficient to generate detailed discussion on rare taxa as shown by Ruiz‐González, Niño‐García (6) and Stegen, Fredrickson (7). Nevertheless, we found that TS-SE-B1 exhibited simultaneously high richness and low inverse Simpson index, which was indicative of highly uneven community and high proportion of rare taxa. Therefore, this finding could be used as starting point for further studies exploring rare taxa and their role in biogeochemical processes in peat layer of Mobile Bay. Furthermore, as suggested by Reviewer 2 via recommendation of paper from Rocca, Simonin (3), we added more discussion on the potential of community coalescence between freshwater and marine taxa, where we found a pattern where marine taxa were commonly found in freshwater sites, but not vice versa, due to their high tolerance of salinity range (line 512-528). 

- Lines 432-440: To my knowledge, not much is known about Archaea in subsurface ecosystems, and in particular in subterranean estuaries. The authors should highlight what is known or not about the topic, in order to better highlight the specific contributions of the presented research.

As previously mentioned, we added archaea-related information in the abstract (line 29-31), introduction (line 62-65), result (line 386-395), and discussions (line 545-558, line 602-612). Generally, we found that the dominant archaeal taxa found in TS-SE were commonly found in other reducing environment and associated with methane cycle, which was also supported the result of metabolic prediction of methane in our study. We also found archaea in TS-NE, despite low relative proportion in comparison to TS-SE. Nevertheless, the archaeal taxa identified in TS-NE were potentially contributed to ammonia/methane oxidation in this site based on the metabolic prediction.

- Line 444-445. Please rewrite this sentence to specify more clearly that NO3 is transformed to NH4 during transit through this site, which is then delivered to the Bay

We revised the sentence into “At the TS-SE site, NO3--rich groundwater percolates through an organic-rich/peat sediment layer and is transformed into NH4+ before discharging to Mobile Bay” (line 563-564).

- Line 446, replace 'and it is likely generated' with ', which are likely generated'

We replaced “and it is likely generated” with “which are likely generated”. 

- Line 450: was related

We revised this sentence.

- Line 458: depending on

We revised this sentence.

References 

1. Montiel D, Lamore A, Stewart J, Dimova N. Is Submarine Groundwater Discharge (SGD) Important for the Historical Fish Kills and Harmful Algal Bloom Events of Mobile Bay? Estuaries and Coasts. 2019;42(2):470-93.

2. Montiel D, Lamore AF, Stewart J, Lambert WJ, Honeck J, Lu Y, et al. Natural groundwater nutrient fluxes exceed anthropogenic inputs in an ecologically impacted estuary: lessons learned from Mobile Bay, Alabama. Biogeochemistry. 2019;145(1-2):1-33.

3. Rocca JD, Simonin M, Bernhardt ES, Washburne AD, Wright JP. Rare microbial taxa emerge when communities collide: freshwater and marine microbiome responses to experimental mixing. Ecology. 2020;101(3):e02956.

4. Héry M, Volant A, Garing C, Luquot L, Elbaz Poulichet F, Gouze P. Diversity and geochemical structuring of bacterial communities along a salinity gradient in a carbonate aquifer subject to seawater intrusion. FEMS microbiology ecology. 2014;90(3):922-34.

5. Wu QL, Zwart G, Schauer M, Kamst-van Agterveld MP, Hahn MW. Bacterioplankton community composition along a salinity gradient of sixteen high-mountain lakes located on the Tibetan Plateau, China. Appl Environ Microbiol. 2006;72(8):5478-85.

6. Ruiz‐González C, Niño‐García JP, del Giorgio PA. Terrestrial origin of bacterial communities in complex boreal freshwater networks. Ecology letters. 2015;18(11):1198-206.

7. Stegen JC, Fredrickson JK, Wilkins MJ, Konopka AE, Nelson WC, Arntzen EV, et al. Groundwater–surface water mixing shifts ecological assembly processes and stimulates organic carbon turnover. Nature communications. 2016;7:11237.

---

## [Editor Report · Decision Letter 1]

11 Jun 2020

Microbial community composition across a coastal hydrological system affected by submarine groundwater discharge (SGD)

PONE-D-20-05547R1

Dear Dr. Adyasari,

We’re pleased to inform you that your manuscript has been judged scientifically suitable for publication and will be formally accepted for publication once it meets all outstanding technical requirements.

Kind regards,

John M. Senko

Academic Editor

PLOS ONE
---

## [Editor Report · Acceptance letter]

17 Jun 2020

PONE-D-20-05547R1 

Microbial community composition across a coastal hydrological system affected by submarine groundwater discharge (SGD) 

Dear Dr. Adyasari:

I'm pleased to inform you that your manuscript has been deemed suitable for publication in PLOS ONE. Congratulations! Your manuscript is now with our production department. 

Kind regards, 

on behalf of

Dr. John M. Senko 

Academic Editor

PLOS ONE